# Integration of affective cues in context-rich and dynamic scenes varies across individuals

Jefferson Ortega [1] ✉, Yuki Murai [2] & David Whitney [1,3,4]

Humans need to make rapid and accurate judgments of others' emotions to understand and navigate the social world around them. To do so, humans combine multiple sources of emotional information from facial expressions and contextual information. However, it is not well understood how different sources of information are integrated, let alone how observers assess which signals should be combined. Across three studies (n = 944) using data from new and previously collected datasets, we investigate whether affective inferences follow a Bayesian framework where information is optimally weighted based on its ambiguity and then combined. We compare this model to a more parsimonious Heuristic integration model that averages cues without considering cue ambiguity. We find that the Bayesian model best predicts individual observers' inferences of affect, but there are significant individual differences in integration strategies, with some individual observers adopting a Heuristic strategy. We also find that integration models that use stable weights instead of dynamic weights, as well as non-integration models, fail to predict observers' affective judgments. Our findings suggest that there are significant idiosyncratic differences in how humans combine affective cues, where some observers use a Bayesian framework to weigh individual cues before integration, while others use efficient but less optimal strategies.

Affect recognition is important for navigating and understanding the social world around us. Humans frequently make rapid and dynamic inferences about the affect of others[1–5], all while simultaneously taking into consideration multiple complex social cues that are readily available to us. When making inferences about others' affect, humans use not only information from facial expressions but also affective information that is retrieved from contextual cues (e.g. body language, tone of voice, setting, other faces, etc.)[6]. The role of contextual information has often been disregarded in early theories of emotion recognition[7,8], however, its importance is increasingly appreciated in both psychological research[9,10] and affective computing[11–14]. When tasked with inferring the emotion of faces in the presence of contextual cues, previous studies have found that face and context cues are automatically and effortlessly integrated[15] and that context can

even override facial information[16,17]. Additionally, the amount of affective information present in the context and in facial information about a person's current affective state is similar[1,3]. Thus, to accurately infer the affect or emotions of others, humans must combine information from both facial expressions and contextual information, quickly and efficiently. Currently, it is unknown how exactly humans combine this information into coherent and reliable judgments of affect.

Previous studies have suggested that humans may combine multiple cues of emotional information into a unified percept by using a Bayesian framework[18–20], which weighs the value of each cue based on its ambiguity. More ambiguous—less reliable—cues are weighted less than unambiguous cues and are then combined[21–23]. Using this framework, our perception of the current emotional state of another

[1]Department of Psychology, University of California, Berkeley, CA, USA. [2]Center for Information and Neural Networks, National Institute of Information and Communications Technology, Osaka, Japan. [3]Helen Wills Neuroscience Institute, University of California, Berkeley, CA, USA. [4]Vision Science Group, University of California, Berkeley, CA, USA. ✉e-mail: Jefferson_ortega@berkeley.edu

person is a combination of the emotional information perceived in a person's face and in the context. For example, if a person currently shows a neutral expression, then the probability of them being sad, P(sad|neutral expression) is low. However, if we observe this person at a funeral then our perception of the person's emotional state is no longer solely dependent on their facial expression. Attending funerals often produces sadness such that P(sad|funeral) is high. By combining these two sources of information, based on their probability of occurring, one would infer that this person is indeed sad even though they have a neutral expression. Although intuitively appealing, recent papers have tested the hypothesis that emotion perception functions under a Bayesian framework and have led to mixed findings, with some studies finding that a Bayesian integration model best captures human observers' emotional judgments[19,24] while other studies have not found this[25].

It is possible that the mixed findings in previous studies are a result of the lack of rich context and/or the absence of dynamic information. Previous studies used unnatural artificial faces[19], non-pictorial descriptions of context[25], and/or static stimuli[24,25]. Each of these limiting factors fails to capture the complex nature of emotion perception as it occurs in more realistic scenes and social interactions. Visual understanding of emotion depends critically on scene context and the dynamic nature of the visual information[1,6,26]. Accordingly, many recent studies have begun to use more naturalistic approaches when investigating human social cognition that capture the context-rich and dynamic nature of social perception[1–4,26–29], however, much of the work on computational social cognition lags behind. Any effort to understand how the brain integrates information for emotion recognition must take into account the dynamic- and context-dependence of emotional information.

In the current study, we address these shortcomings by modeling human observers' continuous affective judgments of real people in natural videos. We aim to investigate whether humans combine affective information using a Bayesian framework, which weighs cues based on their ambiguity. We investigated both static and dynamic stimuli retrieved from videos, which capture much of the visual and contextual complexity that humans experience when perceiving the affect of others in day-to-day social interactions. If humans do indeed use a Bayesian integration framework when combining social cues, then we should find that a Bayesian integration model that weighs cues based on their ambiguity should accurately capture human observers' static and dynamic judgments of human affect.

## Results

### Static integration of context and face/body affective cues is simple but efficient

In Experiment 1, we investigated if spatial context (background scene information) and face/body information are integrated optimally using a Bayesian framework using the Inferential Emotion Tracking (IET) task (Fig. 1). Although there are conflicting reports in the literature, some

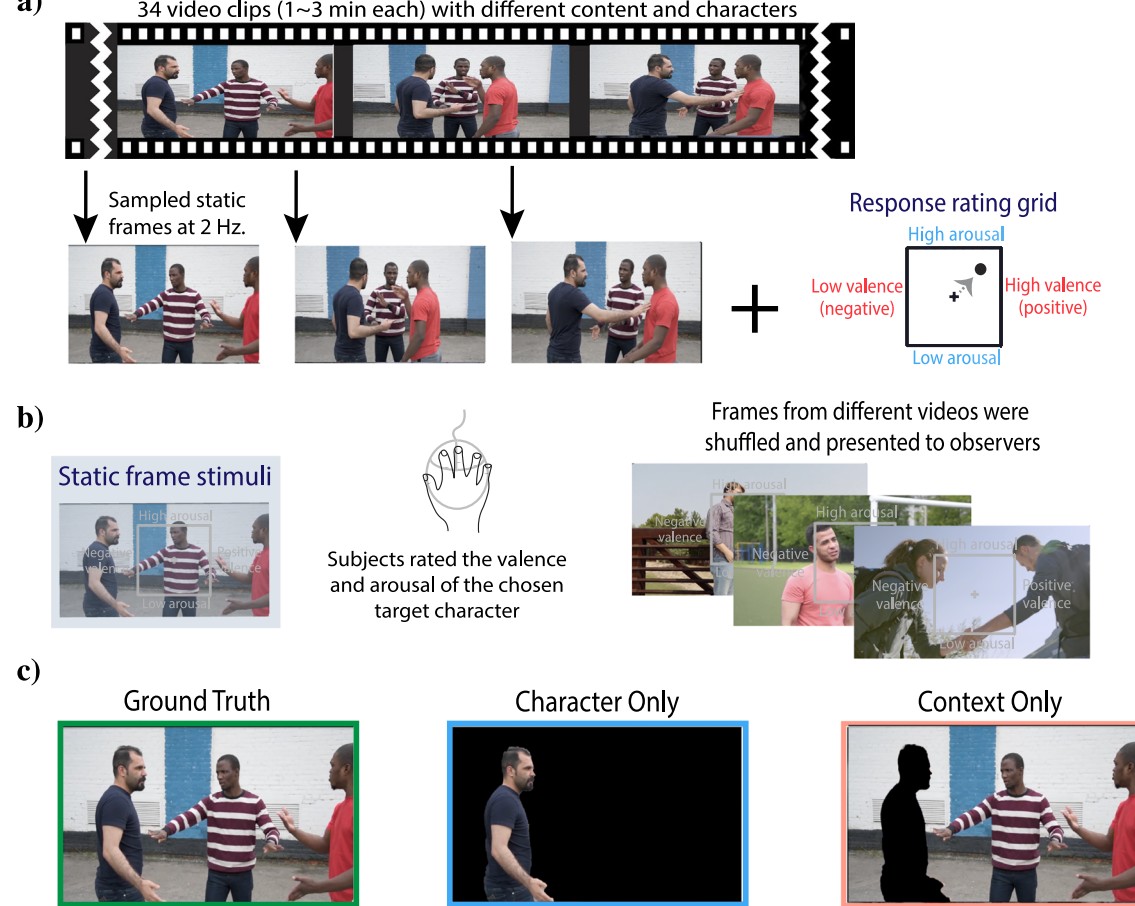

**Fig. 1 | Experiment 1 task design. a** We sampled frames from 34 video clips at 2 Hz (500 ms) and overlaid a 2D valence-arousal grid on each frame (4057 total static frames). **b** Static frames from videos were presented in a random order for each observer. Participants were instructed to rate the valence and arousal of the target character, using a mouse click, and then confirmed their rating with a button located to the left of the video frame. Participants completed as many trials as they could (self-paced) within 1.5 h, for a total of up to 1000 trials per observer. **c** Participants completed the task in either the ground truth, character-only (face/body), or context-only (background scene) condition (see Experiment 1 methods). Screenshots provided by users Aghyad Najjar, u_ctu1ml4mvv, and Joey Velasquevia via Pixabay.

previous work has suggested optimal integration of emotion[19,24]. However, previous studies did not include a competing model to compare with the Bayesian integration model. We therefore introduce the Heuristic model, which combines cues using equal weights, as well as the standard Bayesian model (Fig. 2). We also include three additional competing models to test against the Bayesian integration. We start our investigation using static stimuli in Experiment 1, and then build on this in Experiments 2 and 3 using continuous judgments of affect in dynamic videos of people with rich background context.

Figure 3a shows empirical ratings of the Bayes, Heuristic, and individual cues compared to the ground truth ratings of the static frames in Experiment 1. To compare model performance in Experiment 1, we bootstrapped parameter estimates and 95% confidence intervals. By measuring the Pearson correlation between each model

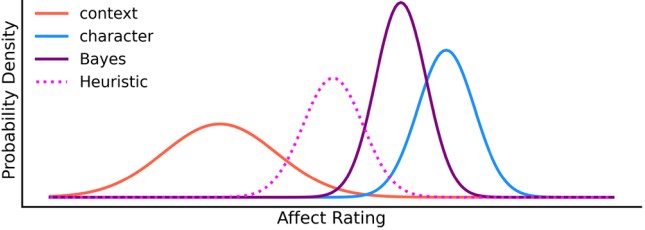

and the ground truth ratings, we found that the Bayes (Pearson r: 0.872, 95% CI: [0.867, 0.877]) and Heuristic (Pearson r: 0.895, 95% CI: [0.892, 0.899]) models outperformed the individual context cues (Pearson r: 0.742, 95% CI: [0.732, 0.751]) and character cues (Pearson r: 0.790, 95% CI: [0.782, 0.798]) (Fig. 3b). Calculating root mean square error (RMSE) led to similar findings with the Bayes (RMSE: 0.125, 95% CI: [0.123, 0.127]) and Heuristic (RMSE: 0.116, 95% CI: [0.114, 0.118]) models having much smaller RMSEs than the individual context (RMSE: 0.173, 95% CI: [0.170, 0.176]) and character (RMSE: 0.177, 95% CI: [0.174, 0.180]) cues (Fig. 3c). This is not surprising, as it replicates prior work[19,24]. Interestingly, comparing the Bayes and Heuristic model using Akaike information criterion (AIC) to control for the number of parameters in each model shows that the (simpler) Heuristic (AIC: −11,858, 95% CI: [−12,123, −11608]) model outperformed the Bayes model (AIC: −10,690, 95% CI: [−10,964, −10,418]) and the individual context (AIC: −5406, 95% CI: [−5684, −5141]) and character (AIC: −5031, 95% CI: [−5328, −4749]) cues (Fig. 3d). These results suggest that observers combine face and context cues when making affect inferences, but they may not weigh the cues based on their ambiguity.

**Dynamic Integration of context and face/body affective cues is simple but efficient**

The results of Experiment 1 indicated that affective cue integration occurs, but the integration process is not a strictly Bayesian-optimal process. One limitation of this experiment—and prior work on the integration of emotion[19,24]—is that the images were static. In Experiment 2, we therefore aimed to replicate our results from Experiment 1 but also to investigate how human observers combine cues dynamically while they are watching a video. Instead of rating the affect of characters in static frames, observers continuously tracked the affect of the target character in a clip using a 2D valence/arousal rating grid that was superimposed on top of the video (Fig. 4a). This experiment addresses not only the shortcomings of our previous experiment but

**Fig. 2 | Bayesian and Heuristic model illustration.** The Bayesian integration of context (red distribution) and character (blue distribution) cues is dependent on the weights assigned to each of the cues. The resulting Bayesian integration of the cues (purple distribution) is biased towards the more reliable (less variance) character cue, and the distribution is narrower than either individual character or context cue. In contrast, the Heuristic model integrates context and character cues while disregarding cue reliability. The resulting Heuristic integration (magenta distribution) assumes equal variance (i.e. equal ambiguity) for both the context and character cues, leading to a distribution positioned between the context and the character cue distributions.

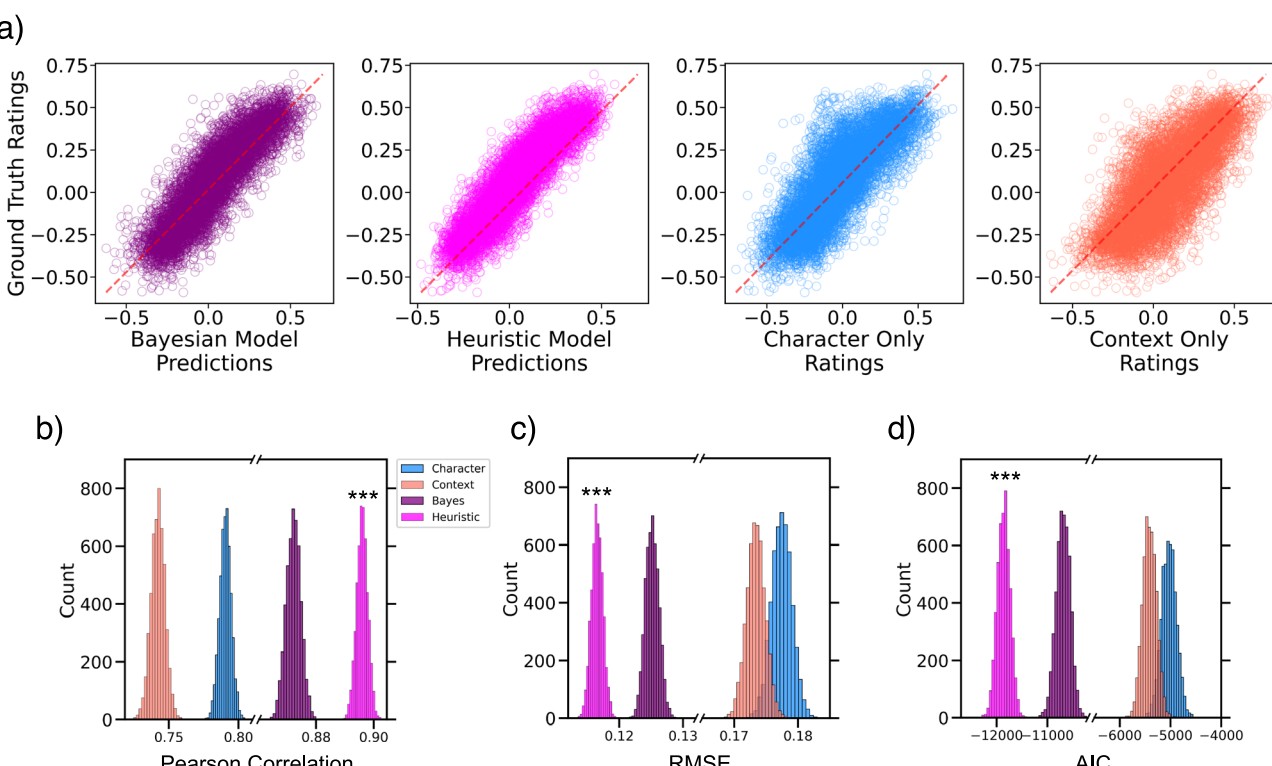

**Fig. 3 | Model performance in Experiment 1 (N = 593). a** Empirical ratings of all models plotted against the ground truth ratings. **b** Pearson correlation with ground truth across models. **c** Root-mean-squared-error across models. **d** Model

comparison using AIC reveals that the Heuristic model outperforms the Bayesian model and both individual cues (Character-only and Context-only) even while disregarding cue weights. Asterisks indicate best performing model.

also limitations in previous literature, as no other study, to the best of our knowledge, has investigated Bayesian cue integration of affect while using continuous ratings.

In Experiment 2, we investigated whether the Bayesian model could accurately predict observers' continuous judgments of characters' affect while they watched a dynamic video. Figure 4b shows an

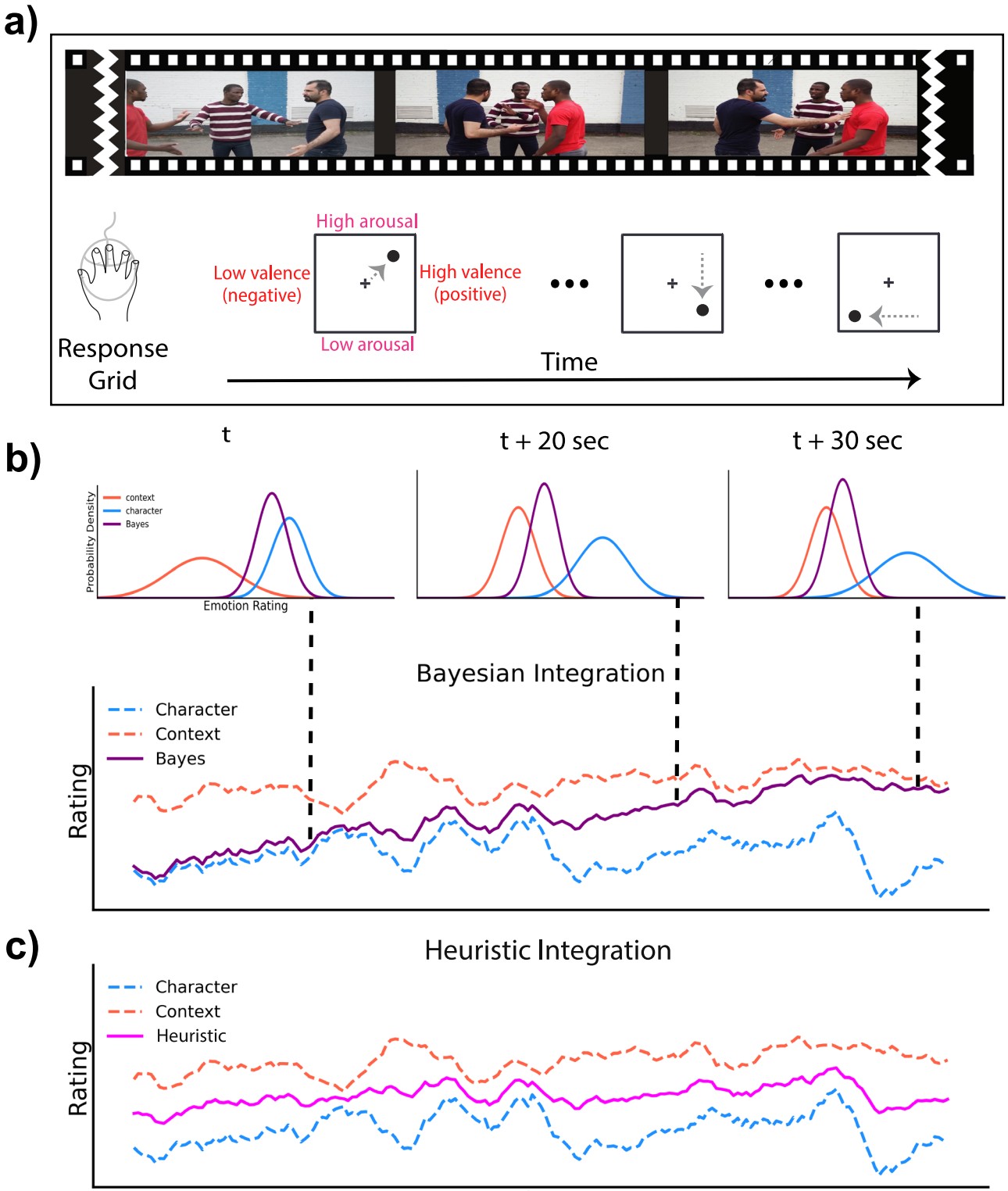

**Fig. 4 | Experiment 2 continuous tracking task and modeling. a** In Experiment 2, participants tracked and rated the valence and arousal of a target character in a video continuously by using a 2D valence-arousal rating grid that was superimposed on top of the video. **b** To model observers' affect judgments in Experiment 2, the same modeling procedure in Experiment 1 was applied at every time point of the video clip as the affect rating evolved. This allows for a dynamic shifting of weights between both the context and character cues, as shown from timepoint *t* to timepoint *t* + 30 s. **c** The Heuristic model computed the average of the consensus of observers' character and context only ratings for all timepoints, similar to Experiment 1. The weights assigned to both the context and character cues remained fixed (50/50) as the affect rating evolved. Screenshots provided by user Aghyad Najjar via Pixabay.

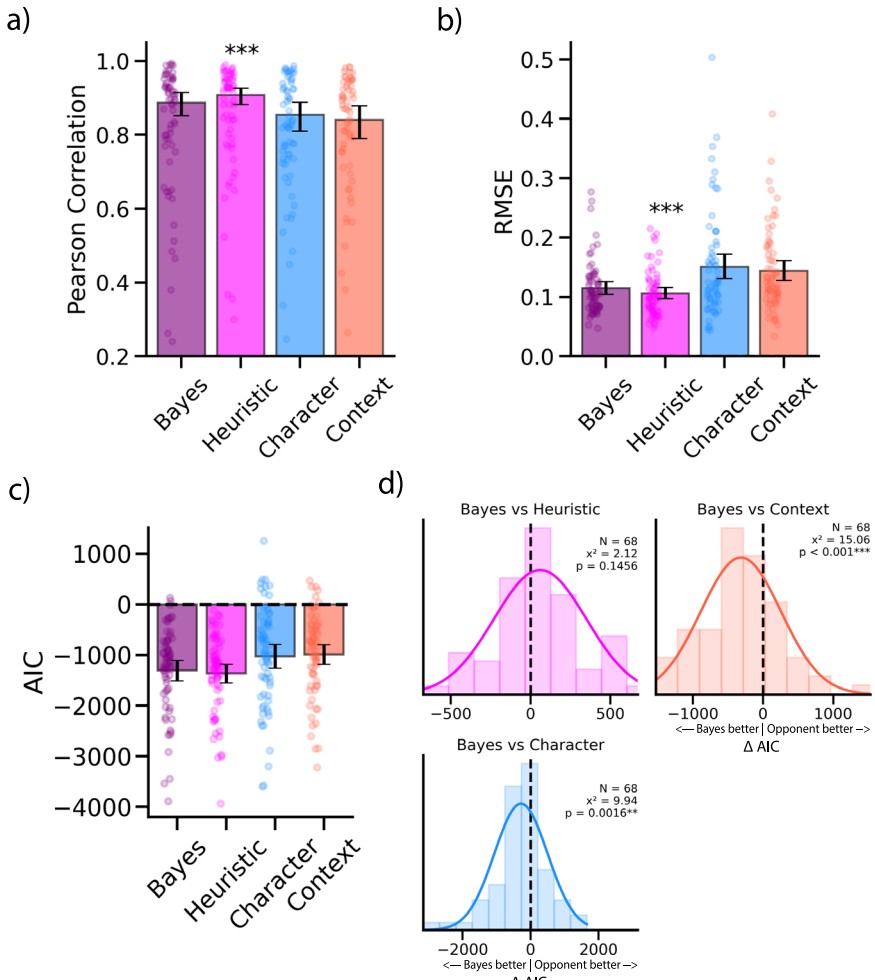

**Fig. 5 | Experiments 2 model performance and comparison (N = 227). a** Pearson correlation with ground truth across models. **b** Root-mean-squared-error across models. **c** Model comparison using AIC reveals that the Heuristic model performs as well as the Bayesian model and outperforms each individual cue. **d** Within-subject AIC differences were calculated between the Bayes model and all other tested models (Bayes AIC - Opponent model AIC). Difference distributions shifted towards negative values indicate that the Bayes model outperforms the opponent model. Chi-squared p-values < 0.05 indicate that the distribution is significantly shifted away from 0. This reveals no significant difference in model performance between the Bayes model and the Heuristic model. Asterisks indicate best-performing model, based on within-subject comparisons. Error bars represent bootstrapped between-subject 95% confidence intervals derived from 5000 iterations.

example of the continuous Bayes integration in which the reliability of the context cue increases over time (red histogram narrows over time in Fig. 4b) which leads to the Bayesian model weighing the context cue relatively higher than the character cue over time. On the other hand, the Heuristic model does not consider the ambiguity of the cues and combines the two cues without setting specific weights for each cue (Fig. 4c).

We found that the Bayes integration model (Pearson r: 0.886, 95% CI: [0.851, 0.914]) outperformed the individual context-only cue (Pearson r: 0.839, 95% CI: [0.789, 0.879]; t(67) = 2.19, p = 0.032) but there was no statistically significant difference between the Bayes model and the character-only cue (Pearson r: 0.853, 95% CI: [0.809, 0.888]; t(67) = 1.44, p = 0.155). However, the Heuristic model (Pearson r: 0.906, 95% CI: [0.883, 0.927]) outperformed the Bayes integration model (Pearson r: 0. 886, 95% CI: [0. 851, 0. 914]; t(67) = 2.45, p = 0.017) and the individual context (Pearson r: 0.839, 95% CI: [0.789, 0.879]; t(67) = 4.25, p < 0.001) and character (Pearson r: 0.853, 95% CI: [0.809, 0.888]; t(67) = 3.14, p = 0.003) cues.

When calculating RMSE, we found that the Bayes integration model (RMSE: 0.115, 95% CI: [0.105, 0.126]) had significantly smaller RMSEs than the individual context (RMSE: 0.144, 95% CI: [0.128, 0.161]; t(67) = −3.55, p <.001) and character cues (RMSE: 0.151, 95% CI: [0.131,

0.172]]; t(67) = −4.19, p < 0.001)(Fig. 3c). However, the Heuristic model (RMSE: 0.106, 95% CI: [0.097,0.116]) had significantly smaller RMSEs than the Bayes integration model (RMSE: 0.115, 95% CI: [0.105, 0.126]; t(67) = −2.57, p = 0.012), the individual context (RMSE: 0.144, 95% CI: [0.128, 0.161]; t(67) = −4.95, p < 0.001), and character (RMSE: 0.151, 95% CI: [0.131, 0.172]; t(67) = −5.14, p < 0.001) cues. Calculating AIC revealed that the Bayes integration model (AIC: −1304, 95% CI: [−1509, −1110]) significantly outperformed the individual context (AIC: −988, 95% CI: [−1187, −797]; t(67) = −4.52, p < 0.001) and character (AIC: −1022, 95% CI: [−1265, −789]; t(67) = −2.94, p = 0.004) cues. AIC values show that the Heuristic model (AIC: −1366, 95% CI: [−1556, −1184]) had similar performance as the Bayes integration model (AIC: −1304, 95% CI: [−1509, −1110]; t(67) = −1.71, p = 0.092). We also found that the Heuristic model outperformed the individual context (AIC: −988, 95% CI: [−1187, −797]; t(67) = −5.38, p < 0.001) and character (AIC: −1022, 95% CI: [−1265, −789]; t(67) = −3.81, p < 0.001) cues (Fig. 5c). We further explored the differences in model performance by computing AIC differences distributions between the Bayes model and all other models/cues (Bayes AIC - Opponent model AIC). We find that the Bayes model did not have significantly different AIC values than the Heuristic model (X² = 2.12, p = 0.145; Chi-squared test) further suggesting that they performed similarly (Fig. 5d). However, the Bayes model did

outperformed the individual context cue ($X^2 = 15.06$, $p < 0.001$) and individual character cue ratings ($X^2 = 9.94$, $p = 0.016$). The results from comparing Pearson R and RMSE values further support the findings of Experiment 1, suggesting that observers combine cues when dynamically inferring affect, but observers may not optimally weigh cues based on the moment-to-moment ambiguity of those cues. However, AIC comparisons suggest that the Bayesian model and the Heuristic model had similar performance in predicting observers' affect ratings. This lack of a difference in AIC may be due to individual differences in integration strategies as some observers may combine cues using a Bayesian framework while others do not. To investigate this question, we aimed to model individual observer ratings in Experiment 3.

### Group level variance predictions of ground truth ratings are best predicted by the Heuristic model

One key concept of Bayesian cue integration is that the integrated distribution should have a narrower width compared to the single cue distributions[30]. This is due to the idea that the combination of two cues should be more reliable (i.e. less ambiguous) than individual cues. Thus, we compared the predicted variance of the combined cues for both Bayes and Heuristic models with the actual ground truth rating variances to measure which model best captured the true variance of the combined cues. In Experiment 1, we found that the Heuristic model had larger correlations and smaller RMSE with the ground truth rating variance than the Bayesian integration model (Table S1). However, variance predictions in Experiment 2 showed that the Bayesian model had larger correlations while the Heuristic model had smaller RMSE. These results suggest that variance predictions may differ based on experimental design, as Experiment 1 used static images while Experiment 2 used dynamic videos. Variance predictions could also vary due to individual differences in the estimation of cue variance across observers. These results further motivate the question of whether individual differences in emotional cue integration exist, which we explored in Experiment 3.

### Individual differences in integration strategies when integrating context and face/body cues in dynamic affective judgments

The Heuristic model outperformed the Bayes model when predicting observers' continuous inferences of affect. However, Experiment 2 used group average ratings for each cue and it therefore does not test whether either model can predict individual observer differences in the integration of context and character cues. Thus, in Experiment 3, we investigated whether we could model individual observers' affective inferences by having each observer continuously rate the affect of the target character in all three experimental conditions. This within-subject design addresses whether cue integration occurs at the individual observer level.

Figure 6a illustrates the modeling procedure for the data in Experiment 3. All participants in Experiment 3 dynamically tracked the affect of a target character in all three rating conditions (context-only, character-only, and ground truth). Using both the Bayes and Heuristic models, we modeled observers' ground truth ratings of the target character with their own ratings in the context-only and character-only conditions (Fig. 6a; see Methods).

We found that the Bayes integration model (Pearson r: 0.596, 95% CI: [0.560, 0.629]) outperformed the Heuristic model (Pearson r: 0.589, 95% CI: [0.552, 0.623]; t(114) = 5.64, p <0.001) and the individual context (Pearson r: 0.478, 95% CI: [0.441, 0.515]; t(114) = 7.95, p <0.001) and character (Pearson r: 0.548, 95% CI: [0.513, 0.581]; t(114) = 14.15, p <0.001) cues (Fig. 6b). The Heuristic model (Pearson r: 0.589, 95% CI: [0.552, 0.623]) also had significantly higher correlations than the individual context (Pearson r: 0. 478, 95% CI: [0. 441, 0. 515]; t(113) = 14.12, p <0.001) and character (Pearson r: 0. 548, 95% CI: [0.744, 0.766]; t(113) = 5.98, p = 0.002) cues. Calculating RMSE led to similar findings, with the Bayes integration model (RMSE: 0.387, 95% CI:

[0.359,0.418]) having significantly smaller RMSEs than the Heuristic model (RMSE: 0.397, 95% CI: [0.367,0.43]; t(113) = −4.19, p <0.001) and the individual context (RMSE: 0.5, 95% CI: [0.461,0.542]; t(113) = −7.32, p <0.001) and character (RMSE: 0.443, 95% CI: [0.408,0.48]; t(113) = −6.95, p <0.001) cues (Fig. 6c). The Heuristic model (RMSE: 0.397, 95% CI: [0.367,0.43]) also had smaller RMSEs than the individual context (RMSE: 0.5, 95% CI: [0.461,0.542]; t(113) = −7.57, p <0.001) and character (RMSE: 0.443, 95% CI: [0.408,0.48]; t(113) = −6.04, p <0.001) cues. Calculating AIC values showed similar results, finding that the Bayes integration model (AIC: 1364, 95% CI: [1016, 1721]) outperformed the Heuristic model (AIC: 1424, 95% CI: [1071, 1786]; t(113) = −5.04, p < 0.001) (Fig. 6d). The Bayes integration model also had significantly smaller AICs than the individual context (AIC: 2406, 95% CI: [2062, 2753]; t(113) = −16.61, p < 0.001) and character (AIC: 1923, 95% CI: [1548, 2296]; t(113) = −10.17, p < 0.001) cues. The Heuristic model also had significantly smaller AICs than the individual context (AIC: 2406, 95% CI: [2062, 2753]; t(113) = −8.4, p <.001) and character (AIC: 1923, 95% CI: [1548, 2296]; t(113) = −16.85, p < 0.001) cues. Once again, we computed AIC differences distributions between the Bayes model and all other models/cues (Bayes AIC - Opponent model AIC) which revealed that the Bayes model had significantly lower AIC values than the Heuristic model ($X^2 = 16.98$, p < 0.001; Chi-squared test)(Fig. 6e). The Bayes model also outperformed the individual context cue ($X^2 = 87.72$, p < 0.001) and individual character cue ratings ($X^2 = 64.88$, p < 0.001).

Overall, these results suggest that individual observers combine affective cues using a Bayesian framework where each cue is weighted by its ambiguity before integration. However, we also found that individual observers may adopt other integration strategies, as some observers' data were best predicted by the Heuristic model (Fig. S1). This suggests that different observers may use highly idiosyncratic integration approaches when combining different affective cues.

### Optimal integration is needed to model human judgments of affect

An alternative hypothesis about how human observers infer affect is that they may alternate between the individual cues that are available to them, instead of integrating cues. Previous studies have found that commonly accepted ideas of integration in decision-making were also predicted by non-integration strategies[31]. Thus, one idea is that observers may just rely on the cue that is the least ambiguous at any given moment and disregard all other cues. For example, if a person's facial expression is very ambiguous but the context is not, an observer may solely rely on the information provided in the context, or vice versa. Additionally, observers may choose to use stable weights instead of dynamically shifting their weights based on the ambiguity of each cue. To test this hypothesis, we investigated five alternative models that alternate between the context and the character cues or integrate both cues by using stable weights. The Context75 model integrates cues by weighting the context cue by 75% and the character cue by 25%. The Character75 model does the opposite and integrates cues by weighting the character cue by 75% and the context cue by 25%. The All-In model chooses the cue at any given moment that has the least amount of ambiguity (akin to a winner-take-all model) while the Random and Correlated models alternate between the two cues with either 50% or 10% probability, respectively.

We first compare model performance across integration models. In Experiment 1, we found that the Heuristic model (AIC = −11858, 95% CI: [−12123, −11608]), outperformed both the Context75 (AIC = −9788, 95% CI: [−10036,− 9527]) and the Character75 (AIC = −9461, 95% CI: [−9742,−9174]) models (Fig. 7a). In Experiment 2, all integration models had similar performances and were not significantly different from each other. There was no statistically significant difference between the Bayesian integration model (AIC = −1303, 95% CI: [−1504, −1110]), and the Context75 (AIC = −1295, 95% CI: [−1498, −1101]; t(67) = −0.25, p = 0.803) or the Character75 (AIC = Mean: −1286, 95% CI:

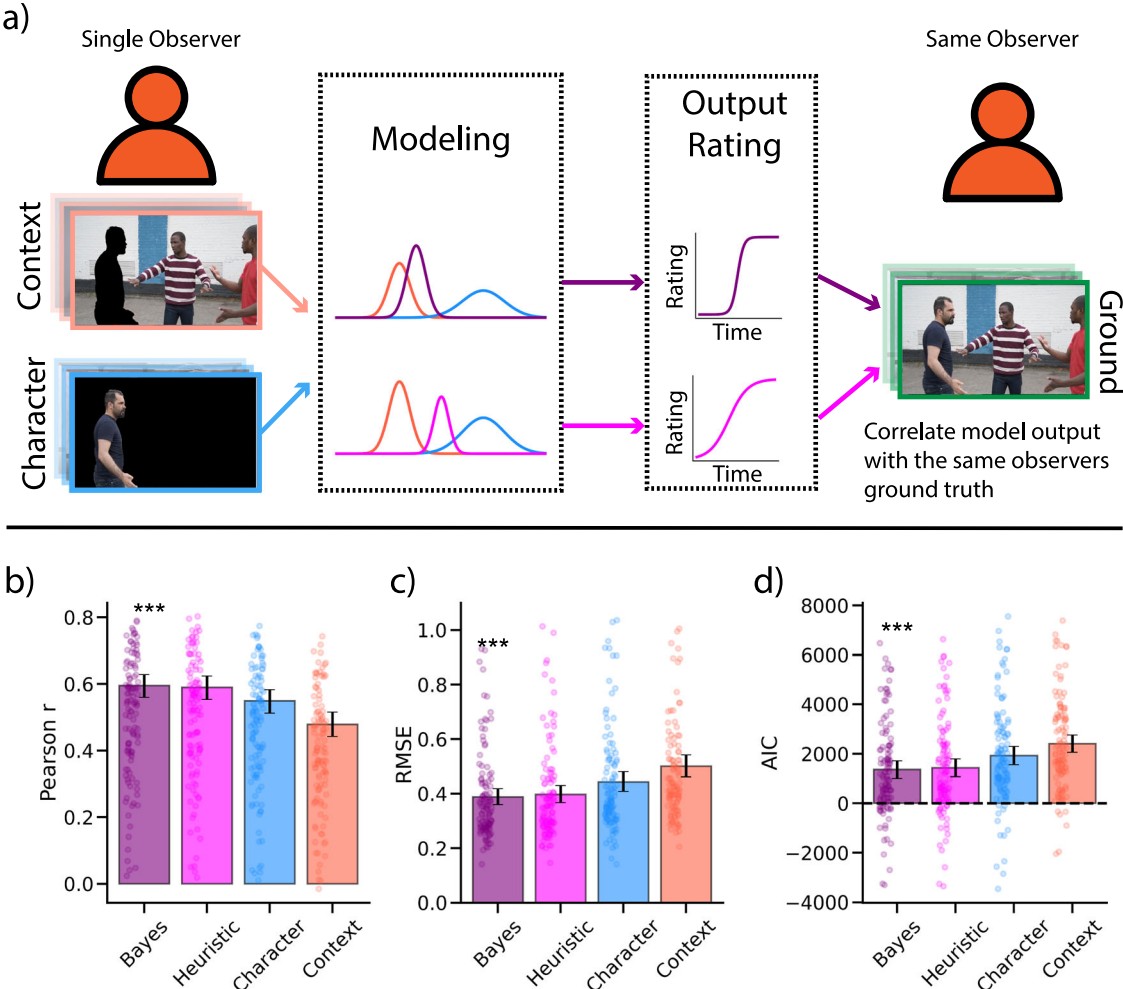

**Fig. 6 | Experiment 3 modeling approach and results (N = 124). a** To investigate whether we could model observers' affective judgments at the individual participant level, observers rated the affect of the target character in all three conditions. A single observer's character and context ratings were then combined to model their own ground truth rating. **b** Pearson correlation with ground truth across models. **c** Root-mean-squared-error across models. **d** Model comparison using AIC reveals that the Bayesian model outperforms the Heuristic model and both individual cues. **e** Within-subject AIC differences distributions between the Bayes model and all other models (Bayes AIC - Opponent model AIC). Chi-squared p-values < 0.05 indicate that the distribution is significantly shifted away from 0. These distributions further support the finding that the Bayes model significantly outperformed the Heuristic model and the individual cues at predicting human observer affective judgements. Asterisks indicate best performing model, based on within-subject comparisons. Error bars represent bootstrapped between-subject 95% confidence intervals derived from 5,000 iterations. Screenshots provided by user Aghyad Najjar via Pixabay.

[−1498, −1084]; t(67) = −0.25, p = 0.803) model. There was also no statistically significant difference between the Heuristic model (AIC = −1365, 95% CI: [−1557,−1183]) and the Context75 (AIC = −1295, 95% CI: [−1498,−1101]; t(67) = −1.42, p = 0.16) or the Character75 (AIC = Mean: −1286, 95% CI: [−1498,−1084]; t(67) = −1.36, p = 0.179) model (Fig. 7b). Computing AIC differences distributions between the Bayes model and all other opponent models revealed similar results (Fig. 7c). In Experiment 3, the Bayesian model (AIC = 1362, 95% CI: [1010, 1720]) had the best performance out of all models, outperforming both the Context75 (AIC = 1776, 95% CI: [1432,2131]; t(113) = −10.73, p < 0.001) and the Character75 (AIC = 1473.485, 95% CI: [1108.501,1839.082]; t(113) = −3.71, p < 0.001) models (Fig. 7d). AIC difference distributions also revealed that the Bayesian model was the best performing model (Fig. 7e). These results suggest that observers use dynamic weights when integrating affective cues and do not rely solely on one cue over the other.

Across all three experiments, we also found that the three alternative non-integration models were less accurate at modeling observers' affective judgments. In Experiment 1, we found that the Bayes integration (AIC = −10,690, 95% CI: [−10,964,−10,418]) and the

Heuristic (AIC = −11858, 95% CI: [−12,123, −11,608]) models had smaller AICs than the All-In (AIC = −5669, 95% CI: [−5955, −5395]), Random Switch (AIC = −5278, 95% CI: [−5548, −5000]), and Correlated Switch models (AIC = −5490, 95% CI: [−5755, −5224])(Fig. 7a). In Experiment 2, the Bayes integration model (Mean: −1303, 95% CI: [−1504,−1110]) had significantly smaller AICs than the All-In (AIC: −997, 95% CI: [−1195,−807]; t(67) = −7.57, p < 0.001), Random Switch (AIC = −849, 95% CI: [−1031,−672]), and Correlated Switch models (AIC: −853, 95% CI: [−1037,−675]; t(67) = −8.14, p < 0.001) (Fig. 7b). The Heuristic model (AIC: −1363, 95% CI: [−1553,−1188]) also had significantly smaller AICs than the All-In (AIC: −997, 95% CI: [−1195,−807]; t(67) = −6.6, p < 0.001), Random Switch (AIC: −849, 95% CI: [−1031,−672]; t(67) = −10.78, p < 0.001), and Correlated Switch models (AIC: −853, 95% CI: [−1037,−675]; t(67) = −10.77, p < 0.001). In Experiment 3, the Bayes integration model (Mean: 1362, 95% CI: [1006,1720]) again had significantly smaller AICs than the All-In (AIC: 2117, 95% CI: [1766,2479]; t(113) = −27.79, p < 0.001), Random Switch (AIC: 2472, 95% CI: [2128,2823]; t(113) = −38.69, p < 0.001), and Correlated Switch models (AIC: 2465, 95% CI: [2119,2823]; t(113) = −38.41, p < 0.001) (Fig. 7d). The Heuristic model (AIC: 1426, 95% CI: [1073,1785]) also had significantly

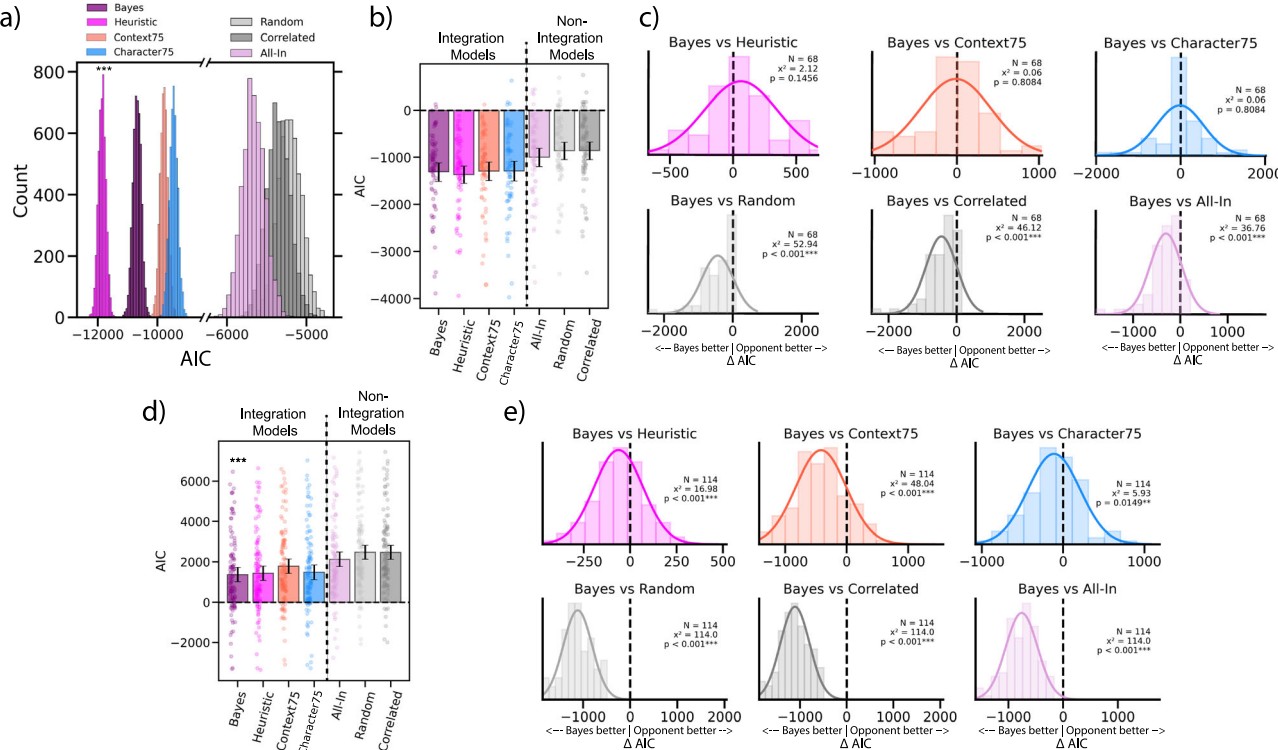

**Fig. 7 | Alternative model comparisons.** We evaluated two alternative integration models with stable weights for context and character cues, as well as three non-integration models, comparing them to the Bayesian and Heuristic models across all three experiments. The Context75 model (red bar) integrated cues by weighting the context cue by 75% and the character cue by 25%. The Character75 (blue bar) model did the opposite and integrated cues by weighting the character cue by 75% and the context cue by 25%. The All-In model (light pink) chose either the context or the character cue, based on which of the two cues was less ambiguous/noisy (lower variance) at each moment (see Methods). The Random model randomly chose (with 50% probability of picking either cue) the context or the character cue at each moment. The Correlated model had a low probability of switching between the character and context cue (10% chance to switch) at each moment. **a** Model comparisons for Experiment 1 (static image stimuli; N = 593). AIC comparison reveals that the Heuristic model had the lowest AIC values and was the best model. **b** Experiment 2 (video stimuli, between-subject; N = 227) model

comparisons. AIC comparison reveals no significant difference between the Bayes and the Heuristic model. **c** Within-subject AIC differences distributions between the Bayes model and all other tested models (Bayes AIC - Opponent model AIC). Chi-squared p-values < 0.05 indicate that the distribution is significantly shifted away from 0. The Bayes model significantly outperformed all non-integration models, but did not have significantly different AIC values than the other integration models. **d** Experiment 3 (video stimuli, within-subject; N = 124) model comparisons. AIC comparisons reveal that the Bayes integration model had significantly lower AIC values than all other models and best predicted individual observers' affective ratings. **e** Within-subject AIC differences distributions for Experiment 3 further support the Bayes model as the best performing model. These results show that optimally integrating the two cues is necessary and that alternating between the two cues or using stable weights is not enough to match observers' affective judgements. Asterisks indicate best performing model. Error bars represent bootstrapped between subject 95% confidence intervals derived from 5000 iterations.

smaller AICs than the All-In (AIC: 2117, 95% CI: [1766,2479]; t(113) = −21.55, p < 0.001), Random Switch (AIC: 2472, 95% CI: [2128,2823]; t(113) = −38.56, p < 0.001), and Correlated Switch models (AIC: 2465, 95% CI: [2119,2823]; t(113) = −38.57, p < 0.001). Computing AIC differences distributions between the Bayes model and all other non-integration models revealed similar results (Fig. 7d). These results reveal that switching between cues is not enough to make accurate predictions of human affective judgments; affective cue integration is needed.

## Discussion

In the present study, we tested the hypothesis that affect perception functions under a Bayesian framework and integrates multiple cues based on their ambiguity. Unlike previous studies, we used stimuli that are dynamic and rich in context, which mimics the complex and continuously evolving environments in which humans make affective inferences. Additionally, we also compared the Bayesian integration model to a Heuristic model to test whether humans optimally weigh cues based on their ambiguity when combining multiple cues. We found that the Heuristic model outperformed the Bayesian integration model when modeling group-level responses. However, we find that the Bayesian model was better at modeling individual observer affect

ratings than the simpler Heuristic model (Figs. 6, S6). Additionally, the Bayesian integration model better predicted observers' affective inferences than individual context or character cues when making inferences on both static and dynamic stimuli at the group and individual observer level. We also found significant individual differences in model performance: some observers' affective inferences were best predicted by the Bayesian model while others were best predicted by the Heuristic model, and these model differences were unique for each observer (Fig. S1). Our findings suggest that humans combine multiple sources of information when making judgments about others' affect, by taking into consideration the ambiguity of each cue. However, integration strategies varied across observers due to individual differences. Finally, we also show that models that only focused on face/body information or just context poorly predicted human judgments of affect; integrating cues from both sources is necessary to make accurate predictions about human affective judgments.

### Modeling emotion integration at the individual level reveals individual differences in integration strategies

Previous results that investigate affective inference using Bayesian Integration models are mixed, with some studies finding that a Bayesian integration model best captures human observers' emotion

judgments[19,24] while other studies do not[25]. In our study, we found that observers differ in the strategies that they use to integrate cues: some observers may use a Bayesian integration framework while others may use more simplistic integration strategies. These results support previous studies that have found individual differences in how observers use and integrate cues[25].

A finer-grained analysis revealed differences in modeling emotional inference at the group level vs the individual level. At the group level, the more parsimonious Heuristic model performed just as well or outperformed the Bayes Integration model by disregarding cue weights and averaging the information in the two cues. The simpler Heuristic integration model also better predicted ground truth rating variance than the Bayesian integration model (Table S1). However, when modeling individual observer ratings, we find that the Bayesian integration model outperformed the Heuristic model (Figs. 6, S1). More interestingly, we find evidence for the existence of significant individual differences in integration strategies among observers, as a significant number of observers had highly idiosyncratic integration strategies (Fig. S1b).

### Individual cues are insufficient to model human affective judgments: optimal cue integration is required

When comparing integration and single-cue, non-integration models, we found that individual cues were not enough to correctly model human observers' affective judgments of the target character. Even switching between cues was ineffective (Fig. 7). Additionally, we found that integrating cues using dynamic weights, instead of stable weights, best predicts observers' ratings, suggesting that observers dynamically update the weights assigned to cues when integrating (Fig. 7). These results suggest that affective cues present in a person's face/body and separately in the context need to be integrated optimally and dynamically to model human affective judgments.

### Influence of stimulus type (static vs. dynamic) on integration

The use of different stimulus types may lead to differences in the integration strategy observers use when inferring affect. In Experiment 1, we found that the Heuristic model outperformed the Bayes integration model. This may be due to the use of static stimuli in Experiment 1, whereas we used dynamic stimuli in Experiments 2 and 3. This suggests that human observers are less likely to consider the ambiguity in different cues when making affective inferences about static stimuli compared to dynamic stimuli. Interestingly, previous studies investigating whether affective inference functions under a Bayesian integration framework have only used static stimuli in their experiments[19,24,25]. While we do not directly compare model performance across experiments, our results highlight the importance of modeling affective inference with more ecologically valid dynamic stimuli and procedures.

### Insights into cognitive mechanisms

Our results point towards the hypothesis that humans may estimate the reliability of different cues in affect perception. This would allow the visual system to make accurate inferences about others' affect by relying on less ambiguous cues while potentially disregarding noisy ones. Thus, the brain can achieve its goals of understanding the social world around it. However, it may also be true that the brain switches from Bayesian to Heuristic integration of cues (and potentially other integration strategies) when needed. Indeed, research in multisensory processing has hypothesized that integration can be flexible and context-dependent, allowing for dynamically adaptive integration mechanisms depending on the information available[32]. Thus, the brain might use different integration strategies when presented with different kinds and amounts of information.

In our study, we used a more complex model, a kernel density estimation model, to fit distributions on observers' ratings, instead of

normal Gaussian distributions. We found that implementing the kernel density estimation model led to worse model performance (Fig. S4). This suggests that while observers may estimate the reliability of cues, they may use more simplistic distributions, like a normal Gaussian, to represent affective information as it could lead to lower computational costs. Alternatively, lower accuracy with the kernel density estimation model could be due to the presence of individual differences. If there are individual differences in affective judgments or even in the estimation of cue ambiguity, then this could lead to worse model accuracy since accuracy is determined at the group-level, and not the individual level for Experiments 1 and 2.

In the multisensory processing literature, it is widely acknowledged that correlation detection is vital to combine different sources of information, especially when they come from different senses[33,34]. Additionally, the brain must solve what is known as the "correspondence problem" where it has to attribute different signals as belonging to the same source while also separating signals that likely belong to different sources. To combine two cues optimally, the brain evaluates their temporal correlation to determine whether they originate from the same source—thereby solving the correspondence problem[35,36]. The correspondence problem is not specific to the multisensory processing field but may also be observed in emotion perception, as well. When making inferences about a person's affect, some information in the context may be related (e.g., being at the funeral of a loved one) or can be independent (e.g., being the custodian at a funeral home). Thus, humans must be able to make accurate judgments of which cues in their environment are related and which are not. A Bayesian integration model for inferring affect may also need to consider that different cues may be processed at different time scales. Previous studies have found optimal models that integrate related multisensory information by temporally filtering the signals and then combining them[37]. Integration for spatial context (e.g., a person's surrounding environment) and temporal context (e.g., chronological narrative information) may also be integrated at different time scales.

### Study limitations

A limitation in our study is that we only used a total of thirty-five video clips, and this cannot capture the endless variations of contexts in which affective processing occurs. Thus, while our study provides an experimental method that mimics real-world affect perception, our analysis does not encompass all possible social environments. It may be the case that the brain does change between multiple integration strategies when inferring affect in different social environments that are not currently captured in our data. Another limitation in our study is that we did not directly measure individual observers' estimates of cue ambiguity and instead used group-level estimates of ambiguity for modeling. Recent work has shown that model performance in the integration of affective ratings is modulated by the types of methods used to measure how observers report their perceived ambiguity of emotional judgments[24]. As we observed in our study, prominent individual differences in affective cue integration exist, and we should expect that this may impact observers' estimates of cue ambiguity uniquely for each individual.

### Suggestions for future research

As mentioned, future research should aim to measure whether individual estimates of cue ambiguity vary across individuals. Our findings also highlight the need to include multiple model comparisons in studies investigating affective cue integration, while also taking into consideration potential individual differences in cue integration. Integration strategies may vary across individuals, and one integration model may not fit all participants' data. This is particularly relevant for atypical and neurodivergent populations, such as those with Autism[4]. Future studies should also investigate and try to differentiate whether cue integration varies for different kinds of context (e.g., spatial vs

temporal context). Finally, researchers should, to the best of their ability, attempt to mimic the conditions in which cognitive functions, like affect processing, occur and should consider the benefits of using more dynamic, continuous, and context-rich stimuli for their experiments.

To conclude, we found that the integration of affective cues in context-rich, dynamic scenes follows a Bayesian framework. However, different observers can use different integration strategies when combining different sources of information, highlighting the presence of idiosyncratic individual differences in emotion perception.

## Methods

### Participants

A total of 944 students at UC Berkeley were recruited for the three experiments. All participants were right-handed with normal or corrected-to-normal vision. In Experiment 1, a total of 593 participants (397 females, 193 males, 3 other) completed the experiment ranging in age from 18 to 33 (M = 22.5, SD = 1.95 years). Data from participants in Experiment 1 was retrieved from Ortega, Chen, & Whitney (2023)[26]. In Experiments 2a and 2b, data was retrieved from Chen & Whitney (2019)[1] which included data from Experiment 2 and Experiment 3 in their study and had a total of 227 participants. In Experiment 3, there was a total of 124 participants (87 females, 34 males, 2 non-binary/non-conforming, 1 prefer not to say) ranging in age from 18 to 40 (M = 21.46, SD = 3.42 years). Ten participants from Experiment 3 were removed from the analysis due to data collection issues resulting from the online website crashing during the experiment before it saved the data. Fifty-two participants completed the task in-lab, and sixty-two participants completed the task online on Pavlovia[38]. Sample size for Experiments 1 and 2 were predetermined by Ortega, Chen, & Whitney (2023) and Chen & Whitney (2019) and were not pre-considered for this study. The sample size for Experiment 3 in this study was decided based on a power analysis calculated using G*Power software to detect a small-medium effect size (r = 0.25) between matched pairs. Results indicated that 101 subjects would be needed to reach a 1 - β = 0.8. We oversampled for an additional 20% to account for potential attrition based on online data collection or inattentive participants, leading to a total sample size of 124.

### Procedure and design

**Experiment 1.** In Experiment 1, we investigated if spatial context (background scene information) and face/body information are integrated optimally using a Bayesian framework. Data from Experiment 1 was collected on a custom-made website. Thirty-five video clips (including Hollywood movies [n = 21], documentaries [n = 2], and home videos [n = 12]) were gathered from an online video-sharing website (YouTube; materials available at https://osf.io/f9rxn/). Frames from the videos were sampled at 2 Hz, resulting in a total of 4057 static frames. Each observer was presented with shuffled frames from different videos presented in a random order, such that visual stimuli in consecutive trials were independent (Fig. 1). Observers completed the Inferential Emotion Tracking[1] (IET) task and were assigned to either the context-only, character-only, or ground-truth condition randomly. Observers in the context-only condition rated the affect of a target character who was blurred out in the frame. Thus, observers in this condition only had access to information in the context to inform their judgments. Observers in the character-only condition rated the affect of a target character who was visible but everything in the context was blurred out. Thus, observers only had access to the information in the target character's face and body. Finally, observers in the ground truth condition rated the affect of a target character in a frame with no edits, so observers had access to both the information available in the context and from the target character. This condition is termed as the "ground truth" because it is what we use to compare model outputs and estimate model performance. In each trial, observers used a two-

dimensional (2D) valence-arousal rating grid to report the valence and arousal of the character in the static image. Valence and arousal ratings ranged in values between −1 and 1, normalized to the size of the 2D bounding box grid. Participants confirmed their response by clicking a "submit" button, which was located on the left of the screen, forcing participants to reset their mouse position after each trial. Participants were allowed to progress through the trials at their own pace and the stimulus frame was presented throughout the duration of the trial. Once participants had confirmed their rating, the next trial began, and a new frame was presented immediately after. Participants had 1.5 hours to complete as many trials as they could, up to a total of 1000 trials (M = 657.35 trials, SD = 363.51).

**Experiment 2.** In Experiment 2, we investigated if a Bayesian integration model could model dynamic affective judgments (see Computational Models Section and Fig. 4). Data analyzed in Experiment 2 was retrieved from Chen and Whitney's (2019)[1] IET task. A total of 34 videos were used by Chen and Whitney (2019) (materials available at https://osf.io/f9rxn). The videos consist of short 1–3 min silent clips (no audio) from various media (Hollywood movies, home videos, and documentaries) containing single or multiple characters. Experiment 2a consisted of 12 video clips retrieved only from Hollywood movies with multiple interacting characters. Experiment 2b consisted of 22 videos from multiple media sources including Hollywood movies, home videos, or documentaries containing either single or multiple characters. Participants used a 2D valence-arousal rating grid to report the perceived valence and arousal of the target character continuously, in real-time (Fig. 4a). Using a between-subjects design, observers were assigned to either the context-only, character-only, or ground-truth condition randomly.

**Experiment 3.** In Experiment 3, we investigated if the Bayesian integration model could predict individual observers' dynamic judgments of affect in the ground truth condition of the IET task using their own perception of affect present in the spatial context (background scene information) and face/body information. Unlike Experiments 1 and 2, the data collected for Experiment 3 were novel and only used in this study. The experimental procedure was identical to Experiment 2 except that this was a within-subject design: each observer rated the target character in each of the three conditions, including character-only, context-only, and ground truth. This was essential in order to use each observer's own character-only and context-only ratings to model their own ground-truth ratings (Fig. 6a). Experiment 3 consisted of 12 silent video clips (no audio) retrieved only from Hollywood movies with multiple interacting characters. Each participant rated the target character in all three conditions in a single session and the order of the conditions in which observers rated the target character was counterbalanced across observers.

### Data analysis

In all experiments, we attempted to model human observers' ground truth affective ratings using the ratings from the individual context and character cues. Participants reported the affect of the target character using valence and arousal dimensions. Valence indicates whether the emotion is negative or positive while arousal indicates whether the emotion is a high-intensity or low-intensity emotion. Across all analyses, model performance for valence and arousal is grouped and averaged; however, model performance for each dimension is provided (Fig. S2). Additionally, correlations between context and character ratings are also provided (Fig. S3).

In Experiment 1, we used the average valence and arousal ratings across observers for the character-only and context-only ratings to model the average valence and arousal ratings of a different group of observers in the ground truth condition. The mean and standard deviation of the ratings were used to create normal probability

distributions for each cue which was used for the Bayes integration model. We then compared the Bayes and Heuristic model outputs to the empirical ratings of the ground truth ratings (Fig. 3a). To compute model performance in Experiment 1, we bootstrapped parameter estimates and 95% confidence intervals. This was done by bootstrapping the data (with 5000 iterations) for each model (shown in Fig. 3a) and calculating the performance metrics (done for Pearson r, root mean square error, and AIC) for each bootstrap.

In Experiment 2, we modeled observers' judgments while they continuously rated the affect of a target character. Ratings were modeled every 100 ms, which was the sampling rate of observers' ratings for Experiment 2. We used the average valence and arousal ratings across observers for the character-only and context-only ratings to model the average valence and arousal ratings of observers in the ground truth condition. This was done for each of the 34 videos used in the experiment. The mean and standard deviation of the ratings were used to create normal probability distributions for each cue which was used for the Bayes integration model. We then compared the Bayes and Heuristic model outputs to the empirical ratings of the ground truth ratings. In Experiments 2 and 3, we also report bootstrapped parameter estimates and 95% confidence intervals for assessing model performance. However, for ease of comparison, we additionally estimate and report t-statistics, bearing in mind the hidden parametric assumptions that may not be satisfied. Computing non-parametric Wilcoxon signed-rank tests, instead of paired t-tests, does not change any of our results. All computed t-tests were two-tailed tests.

In Experiment 3, we used each individual observer's valence and arousal ratings for the character-only and context-only ratings to model their own valence and arousal ratings of the ground truth condition (see Fig. 6a). Ratings were modeled every 40 ms which was the sampling rate of observers' ratings for Experiment 3. This was done for each of the 12 videos used in the experiment. We used the observer's empirical rating at each time point as the mean of the normal probability distribution for each cue. We also used the standard deviation of the ratings across all observers for the normal probability distribution of each cue. We then compared the Bayes and Heuristic model outputs to the empirical ratings of the ground truth condition. In addition to comparing Pearson r, RMSE, and AIC values across models, we also compared model performance in Experiment 3 using protected exceedance probabilities (PXP) which is a measure of how likely any given model is compared with all other models (Fig. S6).

In order to compare the predicted variance of the Bayesian model to the ground truth variance, we randomly sampled from the predicted Bayesian cue integration distribution 1000 times for each trial (Experiment 1) or timepoint (Experiment 2). The predicted variance of the Heuristic model was calculated by averaging the variance of the individual context and character cues for each trial (Experiment 1) or timepoint (Experiment 2).

### Computational models

To model observers' judgements of affect, we used Ong and colleagues (2015) original computational model:

$$P(a,c,f) \propto \frac{P(a|c)P(a|f)}{P(a)} \qquad (1)$$

which posits that the Bayesian integration of the context cue $c$ and face/body cue $f$ is $P(a|c,f)$, which is proportional to the product of the individual cue likelihoods $P(a|c)$ and $P(a|f)$, normalized by the prior probability of the affect occurring $P(a)$. An illustration of the Bayesian integration model is shown in Fig. 2. We calculated the log-sum-exp function, which computes the log sum of probabilities when the log probabilities are available[39]. This allows for normalizing $P(a|c)P(a|f)$

by subtracting the sum of the log probabilities of $P(a|c)$ and $P(a|f)$ instead of dividing by $P(a)$. The log-sum-exp (LSE) is defined as:

$$LSE(x) = \log\left(\sum_{i=1}^{n} e^{x_i - argmax(x)}\right) argmax(x) \qquad (2)$$

where x is the sum of the log probabilities of $P(a|c)$ and $P(a|f)$. The log-sum-exp is computed by calculating the sum of the log probabilities of $P(a|c)$ and $P(a|f)$ and inputting the result into the LSE function. Finally, the $P(a|c,f)$ is then computed by subtracting the LSE from the sum of log probabilities of $P(a|c)$ and $P(a|f)$:

$$P(a|c,f) \propto \log(P(a|c)) + \log(P(a|f)) - LSE(x) \qquad (3)$$

The Heuristic model is computed by averaging the context rating $r_{ctxt}$ and character rating $r_{char}$ and does not consider the reliability of the cues:

$$Heuristic_{model} = \frac{r_{char} + r_{ctxt}}{2} \qquad (4)$$

An illustration of the Heuristic integration model is shown in Fig. 2.

In order to test whether weights need to be dynamically updated during affective cue integration, we investigated the performance of two integration models with stable weights. The Character75 model overweighted the character cue, while the Context75 model overweighted the context cue. We also tested three alternative simple models to investigate whether integration is needed to accurately model human observers' affective judgments. We tested (1) a Random Switch model, (2) a Correlated Switch model, and (3) an All-In model. For each of the models, the consensus (aggregate of responses) across all observers was used as the context-only and character-only cues. The models were tested on each of the videos separately. The random switch model randomly switched from either the context-only cue or the character-only cue with a 50% chance rate at each time point. The Correlated Switch model switched between cues with a 10% chance rate, leading to fewer switches being made across time. Finally, the All-In model chose the cue with the lowest amount of variance across observers at each time point.

In Experiment 2 and Experiment 3, in order to model observers' continuous judgments of affect, we applied the same modeling approach as in Experiment 1 for the Bayesian and Heuristic models. We calculated the Bayesian integration of cues continuously as the ratings evolved (Fig. 4b). The Heuristic model was a simple average of the character and context cues at every time point (Fig. 4c).

### Ethics statement

Informed consent was obtained from all participants, and all experiments in this study were approved by the UC Berkeley Institutional Review Board. All methods were performed in accordance with relevant guidelines and regulations of the UC Berkeley Institutional Review Board. Participants were affiliates of UC Berkeley and participated in the experiment for course credit. All participants were naive to the purpose of the study.

### Reporting summary

Further information on research design is available in the Nature Portfolio Reporting Summary linked to this article.

## Data availability

All data collected and analyzed for this paper are available at https://doi.org/10.17605/OSF.IO/MRTV5[40]. Data from experiments 1[26] and 2[1] were obtained from previous studies. Data used from those studies in this paper are also available in the OSF link provided.

## Code availability

The code for all analyses that support the findings of this study is available at https://doi.org/10.17605/OSF.IO/MRTV5[40].

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

## Acknowledgements

This work was supported by the National Institutes of Health (J.O.; NIH 5F99NS141343, D.W.; NIH R01CA236793), The BRAIN Foundation (J.O. & D.W.; 057899), and the Japan Society for the Promotion of Science Grants-in-Aid for Scientific Research (Y.M.; KAKENHI-23K17648 and 23K22375). We thank users Aghyad Najjar, u_ctu1ml4mvv, and Joey Velasquez on Pixabay for frames used in Figs. 1, 4, and 6.

## Author contributions

J.O., Y.M., and D.W. contributed to the conception of the work. J.O. collected and analyzed all experimental data. J.O. wrote the manuscript with contributions and revisions from Y.M. and D.W. All individuals who meet the authorship criteria of Nature Portfolio journals have been included as authors of this study.

## Competing interests

The authors declare no competing interests.
