## [Transparent Peer Review file · Nature Communications]

Integration of affective cues in context-rich and dynamic scenes varies across individuals

Corresponding Author: Mr Jefferson Ortega

Version 0:

Reviewer comments:

Reviewer #1

(Remarks to the Author)

In their paper "Integration of emotional cues in context-rich, dynamic scenes is simple but efficient", the authors investigate how people make emotional judgments from combinations of facial expressions and contextual cues, using both static (Expt 1) and dynamic (Expt 2-3) naturalistic stimuli. They tested a Bayesian integration model (which weights cues based on ambiguity) against a simpler Heuristic integration model (which ignores ambiguity by treating each cue as having similar ambiguity), and other non-integration models. They find that the Bayesian model does better than non-integration models (i.e., people integrate), but that the simpler Heuristic model performs better than (Expt 1-2) and equal to (Expt 3) the Bayesian model (i.e., integration is "simple but efficient").

I wanted to first start off by acknowledging that I am honored and pleased to see lots of effort to extend my Bayesian cue integration work, almost ten years later. I will be the first to say that the Bayesian model is likely an insufficient description of what I believe people do (although I believe for different reasons than what the authors address in this paper). I still believe that it's mostly in the right direction, and it has changed the way researchers approach multimodal emotion perception. I am very much open to seeing these ideas expanded upon and refined over the years, and I think the current paper can add some useful discussion to the growing literature. I have some comments below that I hope would strengthen the paper.

The major claim in the paper is that the simpler Heuristic model is a better model than the Bayesian model, and I want to push the authors further on their evidence (and actually I believe that the authors are overclaiming based on their evidence), and on the implications.

On the evidence:

One alternative explanation for the results, which the authors do not seem to have entertained, is that the population-level variance is not well estimated, which could arise due to individual differences. This is especially evident because while in Expt 2, which was done between-subjects with some assumptions (that e.g., population-level variance for various cues is shared across participants; a similar assumption was made in Ong et al 2015), the Heuristic model (slightly) outperformed the Bayesian model, but in Expt 3, which was done within-subjects where the same individual's variances were estimated, the Bayesian model actually performed slightly better than the Heuristic model. (I actually weight the results of Expt 3 more because a within-subjects design is a stronger test of the hypothesis.). If there were individual differences in the variances that people assign to different cues, then averaging over to get population-level variance estimates in Expt 2 should introduce more noise into the Bayesian model (compared to the Heuristic model), and will likely hurt the performance of the Bayesian model. This account with the authors' data cannot adjudicate Bayesian vs. Heuristic, but it speaks more to a measurement challenge. (In the same vein, perhaps the Gaussian assumptions or other assumptions in the model might not be the best either).

There is also supporting evidence for individual level differences in cue integration. For instance, Goel et al. (2024; Ref 25 in the paper, which has since been published in Nat Comms) recently showed that there are individual differences in the way people rely on cues; when individual responses are fit within-subject, most people's responses were best predicted by a Situation-only, non-integration heuristic model, then a cue integration model (then a face-only model). This might stem from individual differences in 'strategies' used (some might default to heuristics, some might spend more time thinking). But perhaps there may also be large individual differences in cue reliance (e.g., some people might be "biased" against faces and have large estimates of ambiguity). This other account relates a little to what the authors discuss in lines 549-554; that

people might flexibly switch between simple heuristic integration to more complex Bayesian integration depending on the information available (or e.g., meta-level processes determining cost-benefit outcomes, e.g., Falk Leider's resource-rational analysis).

The authors chose to have participants respond on a 2D valence and arousal grid, especially in Expt 2 and 3 where participants continually rate while watching a video. This likely introduces a lot of cognitive load, especially with the second dimension of arousal being something that is not as familiar to lay participants (further increasing cognitive load). (As a side note, in work in my lab, we've also considered this choice, and stuck with 1D ratings of valence). I wonder what the results of Expts 1-3 would look like if the authors did the modeling only on the valence ratings, rather than the valence+arousal ratings. Why this matters is because if the authors are arguing that participants may be defaulting to a simpler Heuristic integration strategy because of information processing demands, it could be because this task *is* more demanding than previous tasks. (To clarify, I am totally on board with naturalistic stimuli, that is the direction my lab has taken too, for more ecologically-valid studies, but my worry here is on the continuous 2D rating.)

This is perhaps more nit-picky and dividing into the weeds. The results for Expt 2 seem a bit overclaimed. I'm surprised that the Heuristic model (pearson $r = .917$ [.893, .937]) outperforms the Bayesian model ($r = .914$ [.889, .935], $t(67) = 2.76$, $p = .007$). (The RMSE for the Heuristic model is not reported); and the AIC differences is also very small (Bayes: -1396 [-1590, -1209]; Heuristic: -1402 [-1597, -1218]). Frankly I'm surprised that these differences are statistically significant (.003 on a pearson R with 67 degrees of freedom). (Moreover, Fig. 5a actually seems to show the Bayes bar being higher but the ***s are on the lower, Heuristic bar). I would suggest that the authors take a second look at this analysis. Even if everything checks out, the differences are so small that I might actually say the two models are for practical purposes comparable, rather than the "Heuristic model outperforms the Bayesian model".

A similar "overclaiming" happens in Expt 3, where now the Bayesian model outperforms the Heuristic model (also with very small differences; $\Delta r = .596$ vs $.589$, $\Delta \text{RMSE} = .363$ vs $.369$; and ΔAIC was not significant, 1433 vs 1422.). I might suggest that the authors temper their conclusions based on the actual numbers. (I do believe that it's worth arguing that the Heuristics model is an attractive alternative for a number of different reasons: computational cost, difficulty in estimating variances, etc, but I don't think the evidence suggests that it really outperforms the Bayesian model in Expt 2 and 3.)

Onto theoretical implications:

The authors argue that "humans ... combine emotional cues without considering ambiguity as it may be a more efficient and less costly computation." While parsimony seems to be desirable (the simpler Heuristic model outperforms the Bayesian model), there is ample evidence of Bayesian integration in other cognitive domains (non-emotional perception, learning, ...). Taking a step back, it seems unsatisfactory that the brain might employ (more costly) Bayesian integration exclusively in some domains, and (cheaper) integration strategies exclusively in other domains – this suggests machinery to support domain-specific algorithms that are not easily shared across different domains, increasing inefficiency. Rather it seems more likely that (as suggested as a third alternative in lines 550-554) the brain flexibly switches between more costly processing or cheaper processing depending on the situation context (e.g., availability of information; processing demands; incentives for accuracy), and that such switching happens across various domains (other non-emotional multisensory processing as well as emotional cue processing). Where I'm going with this is that what the authors seem to have shown is that in some tasks (e.g., previous studies), people might be using Bayesian cue integration, but in other tasks (e.g., the current Expts 1-2, and maybe Expt 3), people might be using simpler Heuristic integration. The brain can do both, and uses both depending on context (rather than the brain only uses simpler Heuristics for emotional cue integration). This is a more nuanced story than the current framing of the paper (and on a meta-point, is itself a more parsimonious explanation; "the brain uses A and B depending on processing demands" is more parsimonious than "the brain only uses A for emotions and B for other multisensory integration"). This is still generative: it opens up more questions as to the cognitive processing pipeline, meta-level decisions as to when the brain switches strategies, etc etc. I would like to see the authors discuss this more.

Other comments:

In the paragraph beginning on line 562, the authors draw a parallel between their Expt 1 with static images cut from a longer clip, and previous experiments using static stimuli. One big difference though is that the "context" in previous experiments were much richer than in the current Expt 1. (In Ong et al. 2015 it was a game show; Anzellotti et al., 2022, it was winning/losing in tennis games, and Goel et al., 2022 – which I recommend updating to the longer 2024 version, it was a vignette). In the current Expt 1, the "context" is simply the rest of the background scene in the image, less the focal target (Fig. 1). In all the examples in Fig. 1, I actually do not know how the scene contributes to the emotion of the target character (That said, the context-only predictions in Fig. 3 do surprisingly well). In Expts 2 and 3 using dynamic stimuli instead of static stimuli, the biggest difference is *not* that there are changing backgrounds, but that there is additional situational information (conversations, actions, etc...) that participants hear. I think that this is actually a big limitation in comparing Expt 1 to previous studies, and should be discussed.

Overall the writing can be tightened up and proofed. For instance, the writing is sometimes repetitive. Some lines are almost exact repeats, e.g., lines 356-358, 446-448, and 515-517:

"These results suggest that observers do combine face and context cues when making emotion inferences, but they may not weigh the cues based on their ambiguity."

"This suggests that observers do combine face and context cues when making emotion inferences, but they may not weigh each individual cues based on its ambiguity."

On lines 50, 407, 509, the authors use "strategic", e.g., "humans strategically weigh cues based on their ambiguity". Strategic here is not used in a scientific sense (if it is it's not defined), and the use in this context is also not aligned with my lay understanding of that term. There's nothing "strategic" about weighing the most informative cue; it's the optimal (in the mathematically-precise sense) solution. Is a better word for what the authors mean "rational" or "optimal"?

Lines 523-524: the sentence is opposite to what it intends to say.

"These results are at odds with previous findings that have found that a Bayes integration model poorly fits human observers' judgments of emotion"

Previous studies found that the model is a good fit.

Desmond Ong, Ph.D.

(Remarks on code availability)

Reviewer #2

(Remarks to the Author)

The manuscript presents data from three experiments, in which participants rated affective state of individuals in static visual images (Experiment 1) or in dynamic video stimuli (Experiments 2 and 3). The main research question is about how emotional cues from the individual (face/body) and the context (everything else) are integrated to infer their affective state. The authors compared a Bayesian integration model with a simple averaging model (and a number of non-integration models). The results indicate that simple averaging model performed better in representing participants' responses in all datasets. This finding is interpreted as follows: during integration of affective cues to infer others affective states cue ambiguity does not have a substantial influence as simple averaging heuristic could explain responses.

I read the manuscript with great interest. It adopts an interesting methodology to answer the research question. The study design is impressive (blurring the context or the individual to isolate the cues) including many trials of static images and continuous ratings during dynamic stimuli (which I think makes the contribution of the work more important). The results are nicely presented and the interpretations are fair and balanced. However, I have some critical concerns/comments about the methodology and analyses. Given these are addressed or clarified, I think the manuscript would make an important contribution to affective science and social perception fields.

Major comments:

- Almost all stimuli are taken from movies so familiarity with the material is very critical in inferring affective states of the characters. In the manuscript there is no mention of this but it seems like familiarity was assessed at least for Experiment 2 (Chen & Whitney, 2019). It feels like participants who have seen the videos before should be excluded to get a good representation for how various cues are integrated. Because those that have seen the movies, from which the stimuli were taken, have much more information about the context and the character (both at the past and in the future in the timeline).

- One critical question I have is whether the current methodology (data+design+models) is able to adequately differentiate between different models. For instance, if the cues do not differ in ambiguity then the model predictions will not differ greatly (as is the case for at least some of the datasets) and any comparison based on parsimony would favor the simpler averaging model. To assess this, variations in responses from which the distributions are calculated should be presented (perhaps as a supplement). Because it may be problematic, if standard deviations in context-only and character-only conditions are similar across trials. In any case, the authors should show that the models can be identified and distinguished properly (For instance, by presenting a model recovery analysis; see e.g., the section titled "Can you arbitrate between different models?" in Wilson & Collins, 2019. <https://doi.org/10.7554/eLife.49547>).

- Experiment 3 has a within-subject design. So the order effects may be critical for participants' ratings. The authors counterbalanced the orders in which the conditions are presented. But in any case, consecutive ratings may influence each other: the ratings in the third condition (whatever that is) may already involve an integration of stimuli viewed in previous conditions. So it may be difficult to argue about integration of cues in the same way that is done for other experiments. Also, it feels like if participants remembered or used previously acquired information for the later trials, the variation in responses could be smaller reducing the uncertainty used in the Bayesian integration model. If this is the case counterbalancing would shrink uncertainties for every condition which could be in favor of the averaging model since it is a simpler model. I may be wrong or perhaps I misunderstood something but it would be interesting to hear the authors thoughts about this.

- Were the models fit separately for valence and arousal? Or have you tried a multivariate fit/distribution? From Line 241, it sounds like it is the former. In that case, I think separate results should be presented at least as a supplement, so the reader

could see the relative contribution of each dimension. Without this information we do not know whether the superiority of the simple averaging model exists in both dimensions or is it due to for instance better performance on one dimension and a generally inadequate fit of all models in the other.

- I think the authors should also consider a weighted averaging model in which the cues are weighted instead of simply averaged. It would be interesting to see if the weight for the character-only cue is greater than the context for static images (randomly presented) in which there is not much of an evolving context.

- In introduction, discussion and even in abstract, the subject of the study is formulated as the integration of emotional information from facial cues and context. However, the stimuli in character-only condition involve much more than facial expressions. They also include posture, body language, and movement. This is mentioned in passing but I think it should be clearly stated from the abstract what different cues represent. I think this is also one of the main reasons that the results differ from for instance from Goel et al., 2024 (reference 25).

- Please present correlations between context- and character-only ratings as a supplement.

- The methods are not described very clearly. At times I needed to go and read the method sections of Ortega, Chen, Whitney, 2023 & Chen Whitney 2019 to understand the procedure. The explanations of the procedure in Methods section (from Line 184) are confusing. This entire section should be revised to clearly present the procedures. To give a concrete example, the description of Experiment 1 (Line 184-209) does not make it clear that the participants did not view any video stimuli but they instead only rated still images presented randomly. The paragraph starts explaining that participants viewed independent stimuli "Each observer was presented with shuffled frames from different videos presented in a random order" (Line 190). But then it continues "Observers in the context-only condition continuously tracked the emotions of a target character who was blurred out in the video clip." (Line 193). Please go over the entire method section and improve the clarity.

- Comparison for Pearson r at Line 387-388. I was surprised to see that these are different and yet both so close to ceiling. What is the unit of comparison in the t-test with $df=67$?

Minor comments and typos:

- No age and gender distributions are presented for the sample in Experiment 2.
- Line 170-171. "In Experiment 1, there was a total of 593 participants." repetition of the previous sentence.
- Please clearly state whether Experiment 3 is a new data collection for this manuscript
- Please clearly state which experiments are Experiment 2a and 2b in Chen & Whitney, 2019.
- Eq2 (LSE function). What purpose this function fulfills should be explained for the reader?
- Line 294. It should be "when the log probabilities are available"
- No mention of sounds? Were the video stimuli silent?

(Remarks on code availability)

I have not tried to install or run the code. I only looked at the code and it seems that it checks out.

Reviewer #3

(Remarks to the Author)

This paper examines how people integrate emotional information from facial expressions and context across three studies with different dynamics, highlighting a heuristic integration model as the most parsimonious model over a full Bayesian model. This integration model averages cue information without considering cue ambiguity, which makes it less computationally costly. The authors presented their results clearly and while I am interested in understanding how people integrate emotional information, I have major concerns about assumptions from the Bayesian model that might need to be accounted for to properly conduct model comparison. I also would like the authors to investigate potential differences in model fits across emotional video stimuli, something I think that might reveal a potentially interesting and nuanced understanding of how people integrate these cues.

Major concerns

1. My primary concern centers around the assumptions behind the Bayesian integration model. Specifically, the model assumes that each cue is represented with a normal probability distribution (mean and standard deviation). The authors do not present any evidence that the affect ratings are properly represented by a normal distribution and if this assumption is inappropriate, it could cause the Bayesian model to perform poorly. It's quite likely that affect ratings are not always normally distributed; the supplement of one of the published datasets (Chen & Whitney, 2017) shows 2D distributions that do not look like a multivariate normal. I recognize that the authors simply used the formulation from prior research (Ong et al., 2015), which is completely understandable. However, I think in this context where authors want to argue for the superiority of the heuristic model, that the potential violation is highly relevant.

2. A more complete model comparison could include a non-parametric Bayesian model. The authors have a large amount of data, which should make it ideal for estimating the empirical distribution using kernel density estimation or other non-parametric tools. This would allow a non-parametric Bayesian integration model to more properly reflect the empirical priors, rather than collapsing potentially non-parametric distributions into a mean and variance rating. Even if the affect distributions are normal, this modeling approach could simply approximate the normal and offers a more flexible model to account for the

current and future data that extend beyond the situations captured in the emotional video stimuli. The heuristic model effectively just takes the maximum likelihood estimate, making it robust to distribution concerns and is potentially one explanation for why its performance is superior stimuli which might differ in their distributions. This is also could be framed as a positive for the heuristic model since it is more agnostic to distribution.

3. Another major concern is that the authors chose to model valence and arousal separately, without accounting for the covariance between the two. Emotion research has shown a group-level quadratic-like relationship between valence and arousal (Kuppens et al., 2013) and this is particularly important for modeling individual variability in how valence and arousal are related at the individual level. I would like to see the authors apply a joint integration model, especially since the valence-arousal ratings are not independent but averaging across those models effectively treats them as independent. This may be more difficult for the proposed non-parametric Bayesian model, but doable for a multivariate normal Bayesian model.

4. Relatedly, I worry about averaging model performance across valence and arousal. It is understandable for ease of communication, but it's likely the integration of these signals differs. For example, valence is recognized as a main dimension of emotion across cultures while arousal ratings show stronger cultural differences (Jack et al., 2016). Also, a recent large dataset examining valence and arousal ratings shows differences in their distributions even within the same emotion categories (Heffner & FeldmanHall, 2022). Given this I might predict that overall, the valence integration models will fit better than the arousal integration models. It would be helpful if authors separated model performance for valence and arousal, at least to report in the Supplementary materials.

5. I think there's more nuances hidden in the data beyond Bayesian vs Heuristic, and I would love to see the authors present some of the stimuli-specific comparisons. For example, it would be very interesting to see if any of the emotion video stimuli the Bayesian model greatly outperforms the Heuristic model. The Figure 4 visualization intentionally suggests this as the Bayesian integration moves from character to context throughout the video. It was not clear to me how many of the emotional stimuli are shared across the experiments, but assuming there's enough overlap then the authors have an ideal chance to show out of sample generalization (e.g., Bayes model for video 4 beats heuristic model in all three experiments). This could really elevate the understanding of when and why people may want to incorporate the cue ambiguity. Alternatively, if the Heuristic model almost always wins at the emotion-stimuli level this would strengthen the authors conclusions, but if the winning model is inconsistent within stimuli across experiments this would weaken the conclusions from model comparison. It seems plausible that the integration of cue ambiguity might change with the video's context and would fit with a large literature on situation-specific emotion appraisals.

6. The authors should report protected exceedance probabilities (PXP) which is a measure of how likely any given model is compared with all other models in the comparison set (e.g., Piray et al., 2019; Stephan et al., 2009 for original). This also would not require averaging across valence / arousal or emotional stimuli since those could be treated as "subjects" typically are in these modeling papers. This is particularly important and different from comparing lowest average AIC as it considers the relative performance of all models in the set. Because the Bayesian and Heuristic models have similar performance, this measure is ideal to account for model uncertainty and identify the model that is most likely to be the best among the set.

7. Relatedly, I would like to see individual difference measures in Experiment 3 for model fits. For example, a scatter plot of the AIC values for Bayes vs Heuristic model per participant as well as the percentage of subjects who are best fit by either model. One possibility if the data are scattered close to the identity line, it suggests about ~50% are best fit by either model and that participants are essentially indifferent between which model best fits their data. To me, this would suggest that neither model is the "true" model and would offer additional nuance to how people integrate across these cues at the individual level. If this plot reveals clear differences (e.g., 70% of participants are best fit by the Heuristic model), then this strengthens the authors claim for Exp 3 since the Heuristic model ought to be preferred if it fits most participants.

Minor concerns

1. It would be very helpful if authors were to show the empirical distributions for valence and arousal ratings for each emotional video stimuli, this could go in the Supplement.

2. The authors use the term "emotion rating" in many places where "affect rating" is more appropriate. For example, Figure 2 the x-axis is really the affect ratings (valence or arousal). In the computational models, they say they're modeling judgments of emotion, but again valence-arousal ratings are more commonly referred to as affect. I would recommend changing these to distinguish this work from work that is specifically about inferring or judging emotions (i.e., "angry", etc.).

3. Figure 5a. The Pearson correlation bar for Bayes is visually higher than the Heuristic model, but in text the authors report that the Heuristic model has a larger correlation than Bayes (Heuristic r 0.917 and Bayes 0.914, lines 384-390). Please confirm and check.

4. Figure 6a could be clarified. I do not understand the output rating graph, is this meant to reflect the affect rating on the x-axis and the cumulative PDF on the y-axis reported from the model? The authors might want to label the y-axis since the model figure uses a PDF. I expected to see a normal distribution for the output rating.

5. Figure 6a: it might help to have multiple frame examples (i.e., stacked frames) to visually indicate this was for continuous movies.

6. The citations for Goel can be updated into the recent published version (Goel, Jara-Ettinger, Ong, & Gendron, 2024, Nature Communications).

(Remarks on code availability)

I did not review the code line by line but skimmed it to understand how their approach might be able to incorporate a non-

parametric Bayesian model.

I think the authors could include more details in the README or comments of their markdown to make it easier for future readers unfamiliar with these models to follow, but this is also a high bar that is not currently the norm in the field.

Version 1:

Reviewer comments:

Reviewer #1

(Remarks to the Author)

I was Reviewer 1 on the previous round of review. The authors have done a commendable job at answering all my and the other reviewers' comments, including reporting many more analyses.

In general, I'm not surprised by many of the "new changes" in the results, including that (a) the simpler Heuristic model does better for Expt 1 and 2, while (b) the Bayesian model does better at within-subject predictions in Expt 3, and that more observers' behavior are better predicted by the Bayesian than the Heuristic model, but that there exists substantial individual variability. This jives with my intuition, and with previous work (e.g., Goel et al., 2024), and I think this paper makes an exciting contribution to the literature.

I also appreciate that the authors have tempered down their inferences, and I found the additional analyses (e.g., the graphs in the revision letter and supplemental) very helpful in clarifying the nuances in the results.

- I do have a minor question though: for the new KDE-estimation analysis, isn't the normal Gaussian model recoverable by KDE? That is, if the normal Gaussian is really the best performing model, then KDE should be able to recover that normal Gaussian as it's a point in the model space that the KDE will explore. I'm surprised that the KDE does so much worse (comparable to and even worse than the character-only model). I suppose it could be because of the group-level estimation.

I think that this paper is much improved, and I only have minor comments left. I think the writing can be improved in many places, and a thorough proof-read would improve the paper. Here is a non-exhaustive list:

"To conclude, we found that the integration of affective cues in context-rich, dynamic scenes follows a Bayesian framework if individual observer integration is taken into account." – this last clause is unclear... I'd suggest rewriting this.

"Thus, to accurately infer affect, one needs not only facial expressions and body language, but context as well." – the claim in this paragraph-final sentence does not follow from the entire paragraph before. The paragraph is talking about multiple model comparisons and the weightage of different cues.

"Optimal integration is needed to accurately perceive the affect of others" It's difficult to make claims about affect perception "accuracy" because accuracy usually implies some concordance with the "truth". In affective science this is usually taken to be the target's self-report (as we lack better ways of measuring truth), and an observer's perception or inference is said to be accurate iff it matches the target's self-report. In all of the experiments, you do not have the target's actual self-report, so I would argue that accuracy is overclaimed here. Apologies that I didn't catch this in the initial submission. I will note that most of the uses of "accuracy" in the paper is used to mean "model accurately predicting human judgments", which is an accurate use of the term (pun intended), but there are a handful of places where "accurate" (in "accurate affect perception") is imprecise.

"Ground truth" as used in the paper is not used in the commonly used sense of a true "label". In this paper, "ground truth" means "all cues are observed".

(Remarks on code availability)

Reviewer #3

(Remarks to the Author)

I want to thank the authors for their comprehensive response and completing many additional analyses. All of my prior concerns have been addressed and the authors have modified their conclusions regarding the strength of the Bayesian vs Heuristic account accordingly.

After reviewing the revised manuscript, I have no further concerns.

(Remarks on code availability)

Reviewer #4

(Remarks to the Author)

The authors investigate an important question in emotion attribution: how humans integrate information about an agent's face and body with information about the surrounding environment during emotion inferences. They do so using frames from videos, leveraging two innovations: the use of naturalistic stimuli, and dynamic sampling of emotion ratings during the course of a video. The authors investigate integration between face/body and context at the group level, as well as individual differences. The study of individual variation is an important topic that deserves more attention in the field – I praise the authors' effort to engage with this challenging area of research. In my understanding, the analyses rest on a set of assumptions. One of these assumptions – that uncertainty distributions are normal – has been addressed in a response to another reviewer. However, the two other assumptions (discussed below) would need to be tested in order to support firm conclusions about cue integration. I see that the authors already did an impressive amount of work responding to an initial round of reviews, and I believe that it would be useful for the field to have this work in published form. At the same time, the untested assumptions moderate the conclusions that can be derived from the results. At a minimum, these assumptions should be thoroughly discussed in the Discussion section.

Major:

1. According to the methods section "Observers in the context-only condition rated the affect of a target character who was blurred out in the frame.". This suggests that participants were asked to provide a single response for the affect of the person in the video frame, rather than a probability distribution describing how likely the participants thought different affects were. Previous work highlighted the difference between the distribution of ratings across participants and the distribution describing uncertainty within individual participants, finding that the two do not always match. Could this affect the study's conclusions? Individuals who appeared to be better described by the heuristic model might instead be participants whose uncertainty distribution has a minor mode, that was not captured due to asking participants to generate a single affect judgment at each moment rather than asking them to generate a probability distribution that describes their uncertainty. Vice versa, it might also be possible that asking participants to generate a probability distribution describing the likelihood of each possible value of affect (rather than a judgment about the most likely affect) would have revealed more shortcomings of the Bayesian integration approach. The conclusions rest on the assumption that the distribution across participants is equal to the uncertainty within participants. This assumption needs to be tested, or at a minimum this must be acknowledged as a limitation of the study.

2. In Experiment 3, the authors write: "We used the observer's empirical rating at each time point as the mean of the normal probability distribution for each cue. We also used the standard deviation of the ratings across all observers for the normal probability distribution of each cue."

If the probability distribution for each individual observer was modeled using the standard deviation across all observers, this would lead to an incorrect estimate of the distribution of the uncertainty within an individual unless three assumptions are met: first, the assumption already discussed in point 1; second, the assumption that the uncertainty within individuals follows a normal distribution; and third, the assumption that individual differences only affect the mean and not the standard deviation. The issue arising from the second assumption has been addressed using KDE. However, the other two assumptions have not been tested. It is possible I am misunderstanding something, but if my understanding is correct, deriving firm conclusions about cue integration will require testing these assumptions.

3. As noted by Reviewer 1, the word "context" can take on different meanings. In some cases – as in this study – "context" refers to the environment surrounding a target agent. In other cases, "context" is used to refer to situations that lead up to an observed event (for example, "Mark stole a piece of bread after he ran out of food to feed his family.") I read the authors' response to Reviewer 1 on this point, and I still think that context in the sense of "the environment surrounding an agent" and context in the sense of "the representation of the situations leading up to the current frame" are different in general. In some cases, information about the environment surrounding an agent in a video frame can be used to make inferences about the situations that led to the current frame, but this will vary case-by-case. Event understanding and semantic knowledge might play different roles in the analysis of the two types of "context". This does not make the current manuscript less interesting: investigating the role of information about the surrounding environment is valuable as well. However, I concur with Reviewer 1 that this distinction should be made explicit when the study is compared to previous studies that used the term "context" in a different way, because clarifying this will help prevent potential confusion from the readers.

Minor:

4. While the total number of frames is large, the number of videos used is relatively small (34). It is unlikely that the range of scenarios is representative of the variety of situations that occur in daily life. I appreciate that the authors mentioned this in the Discussion section.

(Remarks on code availability)

The code looks clear and straightforward to use to reproduce the results.

Version 2:

Reviewer comments:

Reviewer #1

(Remarks to the Author)

I was Reviewer 1 on the previous rounds. I appreciate the authors' replies, and also their hard work over the past year. I think this paper makes a significant contribution to the literature, and deserves to be published.

(Remarks on code availability)

Reviewer #4

(Remarks to the Author)

The authors have addressed all my concerns.

(Remarks on code availability)

We thank all the reviewers for their constructive feedback. Our original manuscript focused on analyzing our data by investigating model performance based on group performance. However, in light of the reviewers' comments, we have analyzed our data to include more analyses investigating individual differences in model performance across observers. These analyses were done specifically in Experiment 3 where we investigated individual differences in cue integration strategies. We have also fixed an error in our analysis for Experiment 3 which now leads to better Bayesian model performance in Experiment 3. Our new analyses have revealed that while the Heuristic integration model outperforms the Bayesian integration model when using group-averaged data (Experiments 1 and 2), the Bayesian model more accurately predicts *individual* observer data (Experiment 3). In addition, we found that a large majority of observers are best modeled by a Bayesian integration model, however, some observers are best modeled by the Heuristic integration model. These results reveal that different observers may use different integration strategies when combining social cues and highlight the existence of idiosyncratic differences in affective cue integration.

Reviewer #1 (Remarks to the Author):

In their paper “Integration of emotional cues in context-rich, dynamic scenes is simple but efficient“, the authors investigate how people make emotional judgments from combinations of facial expressions and contextual cues, using both static (Expt 1) and dynamic (Expt 2-3) naturalistic stimuli. They tested a Bayesian integration model (which weights cues based on ambiguity) against a simpler Heuristic integration model (which ignores ambiguity by treating each cue as having similar ambiguity), and other non-integration models. They find that the Bayesian model does better than non-integration models (i.e., people integrate), but that the simpler Heuristic model performs better than (Expt 1-2) and equal to (Expt 3) the Bayesian model (i.e., integration is “simple but efficient”).

I wanted to first start off by acknowledging that I am honored and pleased to see lots of effort to extend my Bayesian cue integration work, almost ten years later. I will be the first to say that the Bayesian model is likely an insufficient description of what I believe people do (although I believe for different reasons than what the authors address in this paper). I still believe that it's mostly in the right direction, and it has changed the way researchers approach multimodal emotion perception. I am very much open to seeing these ideas expanded upon and refined over the years, and I think the current paper can add some useful discussion to the growing literature. I have some comments below that I hope would strengthen the paper.

The major claim in the paper is that the simpler Heuristic model is a better model than the Bayesian model, and I want to push the authors further on their evidence (and actually I believe that the authors are overclaiming based on their evidence), and on the implications.

On the evidence:

One alternative explanation for the results, which the authors do not seem to have entertained, is that the population-level variance is not well estimated, which could arise due to individual differences. This is especially evident because while in Expt 2, which was done between-subjects with some assumptions (that e.g., population-level variance for various cues is shared across participants; a similar assumption was made in Ong et al 2015), the Heuristic model (slightly) outperformed the Bayesian model, but in Expt 3, which was done within-subjects where the same individual's variances were estimated, the Bayesian model actually performed slightly better than the Heuristic model. (I actually weight the results of Expt 3 more because a within-subjects design is a stronger test of the hypothesis.). If there were individual differences in the variances that people assign to different cues, then averaging over to get population-level variance estimates in Expt 2 should introduce more noise into the Bayesian model (compared to the Heuristic model), and will likely hurt the performance of the Bayesian model. This account with the authors' data cannot adjudicate Bayesian vs. Heuristic, but it speaks more to a measurement challenge. (In the same vein, perhaps the Gaussian assumptions or other assumptions in the model might not be the best either).

There is also supporting evidence for individual level differences in cue integration. For instance, Goel et al. (2024; Ref 25 in the paper, which has since been published in Nat Comms) recently showed that there are individual differences in the way people rely on cues; when individual responses are fit within-subject, most people's responses were best predicted by a Situation-only, non-integration heuristic model, then a cue integration model (then a face-only model). This might stem from individual differences in 'strategies' used (some might default to heuristics, some might spend more time thinking). But perhaps there may also be large individual differences in cue reliance (e.g., some people might be "biased" against faces and have large estimates of ambiguity). This other account relates a little to what the authors discuss in lines 549-554; that people might flexibly switch between simple heuristic integration to more complex Bayesian integration depending on the information available (or e.g., meta-level processes determining cost-benefit outcomes, e.g., Falk Leider's resource-rational analysis).

Good point; we agree. To address these comments, we examined the individual differences testing whether individual observer's ratings were better captured by the Bayesian model than the Heuristic model as seen below in Figure S1:

Figure S1

We investigated whether there were individual differences in integration strategies by plotting model performance between the Bayesian and Heuristic models for each individual participant. We first computed the differences between the Akaike information criterion (AIC) values of the Bayesian model and the Heuristic model for each participant. Positive values indicate that the participants' data was best captured by the Heuristic model while negative values indicate the Bayesian model best captured their ratings. We then compared the AIC difference scores to a permuted null distribution. The permuted null was computed by shuffling observers' ratings such that a random observer's context and character ratings were integrated to predict each observer's ground truth rating. This allows us to investigate whether there are significant idiosyncratic differences within observers, or if observers' data is interchangeable between observers.

(a) shows an example of the AIC difference scores obtained for a single observer's permutation (gray distribution; 100 iterations) compared to their observed AIC difference score (vertical red dashed line).

(b) shows the same analysis for all observers where the shaded region is the 95% confidence interval (CI) of the permuted null distribution. Observers who had significant AIC differences (those that were outside of the permuted 95% CI) are shown as a yellow diamond and have an asterisk (*) at the top of the figure. In total, there were 30 observers who had significant individual differences which is significantly more than expected by chance ($\chi^2 = 101$, $p < 0.001$). Importantly, this suggests that observers cannot be swapped with other observers, indicating significant individual differences.

(c) We also found that more individual observers were best modeled by the Bayesian model than the Heuristic model ($\chi^2 = 16.98$, $p < 0.001$).

(d) To further explore whether there were significant individual differences within observers, we performed a split-half analysis where we split each observer's AIC difference scores in half, randomly, and compared the average AIC difference score for each half for 10,000 iterations using RMSE (within-subject analysis). We compared this to a between-subject analysis where we randomly selected a pair of observers AIC difference scores and compared the average AIC difference score for 10,000 iterations using RMSE (between-subject analysis). We find that AIC differences are more similar within-subjects than between-subjects, providing further evidence for individual differences among observers ($p = 0.0072$; permutation test).

(e) Finally, we investigated how the performance of the Bayesian model differs if we were to shuffle context and character ratings across observers. We find that integration cues retrieved from a random pair of observers lead to worse model performance than when using each observer's own ratings for character and context cues ($t(113) = -12.72$, $p < 0.001$).

Overall, these findings suggest that different observers may use different integration strategies when combining affect information, highlighting the presence of idiosyncratic differences across observers.

These results have been added as supplementary Figure 1 (Figure S1).

The authors chose to have participants respond on a 2D valence and arousal grid, especially in Expt 2 and 3 where participants continually rate while watching a video. This likely introduces a lot of cognitive load, especially with the second dimension of arousal being something that is not as familiar to lay participants (further increasing cognitive load). (As a side note, in work in my lab, we've also considered this choice, and stuck with 1D ratings of valence). I wonder what the results of Expts 1-3 would look like if the authors did the modeling only on the valence ratings, rather than the valence+arousal ratings. Why this matters is because if the authors are arguing that participants may be defaulting to a simpler Heuristic integration strategy because of information processing demands, it could be because this task *is* more demanding than previous tasks. (To clarify, I am totally on board with naturalistic stimuli, that is the direction my lab has taken too, for more ecologically-valid studies, but my worry here is on the continuous 2D rating.)

Good question. We now provide separate valence and arousal performance for all experiments as supplemental figures (Figure S2).

Figure S2

We find that the Heuristic model outperformed the Bayesian model in both valence and arousal dimensions in Experiment 1. In Experiment 2, we find no difference in performance between the Bayesian and Heuristic models in the valence dimension ($t(67) = 0.52$, $p = 0.603$) nor in the arousal ratings ($t(67) = 1.87$, $p = 0.07$). Finally, we find a significant difference in performance between the Bayesian and Heuristic models when predicting observers' ratings in valence ($t(113) = -6.77$, $p < 0.001$) but not in arousal ($t(113) = -0.44$, $p = 0.662$) dimensions in Experiment 3.

We have added these results in the supplementary information (Figure S2).

This is perhaps more nit-picky and dividing into the weeds. The results for Expt 2 seem a bit overclaimed. I'm surprised that the Heuristic model (pearson $r = .917$ [.893, .937]) outperforms the Bayesian model ($r = .914$ [.889, .935], $t(67) = 2.76$, $p = .007$). (The RMSE for the Heuristic model is not reported); and the AIC differences is also very small (Bayes: -1396 [-1590, -1209]; Heuristic: -1402 [-1597, -1218]). Frankly I'm surprised that these differences are statistically significant (.003 on a pearson R with 67 degrees of freedom). (Moreover, Fig. 5a actually seems to show the Bayes bar being higher but the ***s are on the lower, Heuristic bar). I would suggest that the authors take a second look at this analysis. Even if everything checks out, the

differences are so small that I might actually say the two models are for practical purposes comparable, rather than the “Heuristic model outperforms the Bayesian model”.

A similar “overclaiming” happens in Expt 3, where now the Bayesian model outperforms the Heuristic model (also with very small differences; delta-r = .596 vs .589, delta-RMSE = .363 vs .369; and delta-AIC was not significant, 1433 vs 1422.).

I might suggest that the authors temper their conclusions based on the actual numbers. (I do believe that it’s worth arguing that the Heuristics model is an attractive alternative for a number of different reasons: computational cost, difficulty in estimating variances, etc, but I don’t think the evidence suggests that it really outperforms the Bayesian model in Expt 2 and 3.)

We thank the reviewer for pointing out this discrepancy; we have clarified and corrected the results section. Our original analysis code had an error and we have corrected it. The fixed code does indeed reveal that the Heuristic model is outperforming the Bayesian model, except for when calculating AIC (Figure 5):

We have revised the results section to reflect these results.

Onto theoretical implications:

The authors argue that “humans ... combine emotional cues without considering ambiguity as it may be a more efficient and less costly computation.” While parsimony seems to be desirable (the simpler Heuristic model outperforms the Bayesian model), there is ample evidence of Bayesian integration in other cognitive domains (non-emotional perception, learning, ...). Taking a step back, it seems unsatisfactory that the brain might employ (more costly) Bayesian integration exclusively in some domains, and (cheaper) integration strategies exclusively in other domains – this suggests machinery to support domain-specific algorithms that are not easily shared across different domains, increasing inefficiency. Rather it seems more likely that (as suggested as a third alternative in lines 550-554) the brain flexibly switches between more

costly processing or cheaper processing depending on the situation context (e.g., availability of information; processing demands; incentives for accuracy), and that such switching happens across various domains (other non-emotional multisensory processing as well as emotional cue processing). Where I'm going with this is that what the authors seem to have shown is that in some tasks (e.g., previous studies), people might be using Bayesian cue integration, but in other tasks (e.g., the current Expts 1-2, and maybe Expt 3), people might be using simpler Heuristic integration. The brain can do both, and uses both depending on context (rather than the brain only uses simpler Heuristics for emotional cue integration). This is a more nuanced story than the current framing of the paper (and on a meta-point, is itself a more parsimonious explanation; "the brain uses A and B depending on processing demands" is more parsimonious than "the brain only uses A for emotions and B for other multisensory integration"). This is still generative: it opens up more questions as to the cognitive processing pipeline, meta-level decisions as to when the brain switches strategies, etc etc. I would like to see the authors discuss this more.

Excellent point. We agree with the reviewer that observers may use different strategies depending on the information available to them and that these strategies may also depend on individual biases towards different sources of information (context vs face; e.g. Goel et al (2024)). In light of our new analyses in Experiment 3, we find that the Bayesian integration model best predicts individual observers' affect ratings and outperforms the Heuristic model. We also find significant individual differences in model performance across observers (Figure S1). We have revised the manuscript to reflect our new analyses in Experiment 3 in addition to building upon the idea of different integration strategies in the discussion of the manuscript.

Other comments:

In the paragraph beginning on line 562, the authors draw a parallel between their Expt 1 with static images cut from a longer clip, and previous experiments using static stimuli. One big difference though is that the "context" in previous experiments were much richer than in the current Expt 1. (In Ong et al. 2015 it was a game show; Anzellotti et al., 2022, it was winning/losing in tennis games, and Goel et al., 2022 – which I recommend updating to the longer 2024 version, it was a vignette). In the current Expt 1, the "context" is simply the rest of the background scene in the image, less the focal target (Fig. 1). In all the examples in Fig. 1, I actually do not know how the scene contributes to the emotion of the target character (That said, the context-only predictions in Fig. 3 do surprisingly well). In Expts 2 and 3 using dynamic stimuli instead of static stimuli, the biggest difference is *not* that there are changing backgrounds, but that there is additional situational information (conversations, actions, etc...) that participants hear. I think that this is actually a big limitation in comparing Expt 1 to previous studies, and should be discussed.

We appreciate the reviewer's comment, however, we would like to respectfully push back on what the reviewer considers as "rich" context. The contextual information in our stimuli mimics the natural environments in which context is present when observers

make social inferences in the real world. The reason why visual context is so informative (especially in Exp 1 with static stimuli) is because the environments in which we exist are often a reflection of us and our emotions and feelings, the company we keep, the places we work and live, etc. We do want to add that there is no audio information in our videos. We have revised the manuscript to reflect this information.

In Ong et al. 2015 and Anzellotti et al., 2022, complex social inferences are made during specific social conditions (e.g. how would you feel in this *specific* situation) which limits the generalizability of the results across different contexts and social decisions. The results of Goel et al. (2024) begin to investigate the complexity in which emotions are processed by using various verbal social scenarios in their study. However, our study goes beyond these methods by requiring observers to visually perceive and process the situations in which emotions are expressed without using verbal cues or descriptions. Thus, our stimuli capture the visual complexity of emotions and the contexts in which they are expressed and perceived. This is why we believe that our stimuli are indeed “rich” in context.

Overall the writing can be tightened up and proofed. For instance, the writing is sometimes repetitive. Some lines are almost exact repeats, e.g., lines 356-358, 446-448, and 515-517: “These results suggest that observers do combine face and context cues when making emotion inferences, but they may not weigh the cues based on their ambiguity.” “This suggests that observers do combine face and context cues when making emotion inferences, but they may not weigh each individual cues based on its ambiguity.”

We have revised the writing in the results sections to be less repetitive.

On lines 50, 407, 509, the authors use “strategic”, e.g., “humans strategically weigh cues based on their ambiguity”. Strategic here is not used in a scientific sense (if it is it’s not defined), and the use in this context is also not aligned with my lay understanding of that term. There’s nothing “strategic” about weighing the most informative cue; it’s the optimal (in the mathematically-precise sense) solution. Is a better word for what the authors mean “rational” or “optimal”?

We have changed the word strategic to optimal in our manuscript.

Lines 523-524: the sentence is opposite to what it intends to say. “These results are at odds with previous findings that have found that a Bayes integration model poorly fits human observers’ judgments of emotion”
Previous studies found that the model is a good fit.

This statement was in regard to Goel et al. (2024) who found the situation-only model better fit observers’ responses than the Bayes model. We recognize that “poorly fits” is not exactly correct and have changed this sentence so that it reflects the findings of Goel et al. (2024).

Reviewer #2 (Remarks to the Author):

The manuscript presents data from three experiments, in which participants rated affective state of individuals in static visual images (Experiment 1) or in dynamic video stimuli (Experiments 2 and 3). The main research question is about how emotional cues from the individual (face/body) and the context (everything else) are integrated to infer their affective state. The authors compared a Bayesian integration model with a simple averaging model (and a number of non-integration models). The results indicate that simple averaging model performed better in representing participants' responses in all datasets. This finding is interpreted as follows: during integration of affective cues to infer others affective states cue ambiguity does not have a substantial influence as simple averaging heuristic could explain responses.

I read the manuscript with great interest. It adopts an interesting methodology to answer the research question. The study design is impressive (blurring the context or the individual to isolate the cues) including many trials of static images and continuous ratings during dynamic stimuli (which I think makes the contribution of the work more important). The results are nicely presented and the interpretations are fair and balanced. However, I have some critical concerns/comments about the methodology and analyses. Given these are addressed or clarified, I think the manuscript would make an important contribution to affective science and social perception fields.

Major comments:

- Almost all stimuli are taken from movies so familiarity with the material is very critical in inferring affective states of the characters. In the manuscript there is no mention of this but it seems like familiarity was assessed at least for Experiment 2 (Chen & Whitney, 2019). It feels like participants who have seen the videos before should be excluded to get a good representation for how various cues are integrated. Because those that have seen the movies, from which the stimuli were taken, have much more information about the context and the character (both at the past and in the future in the timeline).

In a previous study in which we retrieved the stimuli of Experiment 2, Chen & Whitney, (2019) investigated whether familiarity with the clips biased observers ratings of the context-only stimuli. They found that familiarity with the clips did not significantly influence their ratings (see 2nd paragraph of the Stimuli section in the Supplementary Information). Additionally, in another study, Chen & Whitney (2021) calculated cross-correlations between the ratings of observers who were familiar and those who were not familiar with the video clips. They found no delay between the two groups of observers

(see Fig. S5 in supplementary information of Chen & Whitney 2021). Together, these results suggest no significant impact of familiarity on the measures here.

- One critical question I have is whether the current methodology (data+design+models) is able to adequately differentiate between different models. For instance, if the cues do not differ in ambiguity then the model predictions will not differ greatly (as is the case for at least some of the datasets) and any comparison based on parsimony would favor the simpler averaging model. To assess this, variations in responses from which the distributions are calculated should be presented (perhaps as a supplement). Because it may be problematic, if standard deviations in context-only and character-only conditions are similar across trials. In any case, the authors should show that the models can be identified and distinguished properly (For instance, by presenting a model recovery analysis; see e.g., the section titled "Can you arbitrate between different models?" in Wilson & Collins, 2019. <https://doi.org/10.7554/eLife.49547>).

We agree with the reviewer that cue variances need to differ in order for the Bayesian model to differ from the Heuristic model. We have plotted the variance of both context and character ratings in the figure below (Figure R1):

Figure R1a shows the context and character variance for a single video across all time points. In this individual video, we can see that context ratings have a larger variance than character ratings at all time points. However, variance does change throughout the video. We can compare this to the same plot of all other videos shown in figure R1b. We can qualitatively observe that the relationship between character-only and context-only

variance differs greatly across all videos, indicating that the variances in both conditions are not similar across videos and the cues do differ in ambiguity.

In regards to the model recovery analysis, Wilson & Collins, (2019) propose that:

“model recovery involves simulating data from all models (with a range of parameter values carefully selected as in the case of parameter recovery) and then fitting that data with all models to determine the extent to which fake data generated from model A is best fit by model A as opposed to model B”

This is not an ideal recovery analysis for our data since the Bayesian model receives the variance of the different cues. Specifically, if we generate fake time-series data for both context-only and character-only cues and combine them using fake weights (i.e. cue variance), the Bayesian model will have access to these weights and will recover the integrated simulated cues every time.

Looking back at the variances above, because the variance of the context-only and character-only cues are so different across videos, they have to generate different data because of the parameters of the models. Additionally, if the two models produced similar results, we would not see an advantage of the Heuristic model in both Experiment 1 and 2 in predicting the ground truth ratings nor the advantage of the Bayesian model in Experiment 3.

Finally, another reviewer suggested computing protected exceedance probabilities (PXP) which is a measure of how likely any given model is compared with all other models in the comparison set. This is another form of model comparison which we use in our revised manuscript. We find that the Bayesian model had higher PXP values than the Heuristic model, indicating that it better fit observers' ratings in Experiment 3 than the Heuristic model. We have added this analysis to the supplemental (Figure S6):

Figure S6

- Experiment 3 has a within-subject design. So the order effects may be critical for participants' ratings. The authors counterbalanced the orders in which the conditions are presented. But in any case, consecutive ratings may influence each other: the ratings in the third condition (whatever that is) may already involve an integration of stimuli viewed in previous conditions. So it may be difficult to argue about integration of cues in the same way that is done for other experiments. Also, it feels like if participants remembered or used previously acquired information for the later trials, the variation in responses could be smaller reducing the uncertainty used in the Bayesian integration model. If this is the case counterbalancing would shrink uncertainties for every condition which could be in favor of the averaging model since it is a simpler model. I may be wrong or perhaps I misunderstood something but it would be interesting to hear the authors thoughts about this.

The within-subject design in Experiment 3 was intentional in order to model individual observers' data instead of the group analysis that was done in Experiment 2. Since we are averaging model performance across all subjects, regardless of stimulus order, any order effects should be averaged out. In regards to variation in responses, we use the variations in the responses across all observers (an assumption made in the analysis) and do not use an individual observer's perceived variance of individual cues. Thus, any differences in cue variance due to order effects should also be averaged out.

In order to investigate whether variance within cues was different across experiments, we looked at the z-scored (within experiment) variance across conditions in both Experiments 2 and 3 while only including ratings of videos that were in both experiments:

Figure R2

In Figure R2, we find that variance is similar in both experiments 2 and 3 within cue but different across cues. This suggests that the variance of each cue did not differ due to the within-subjects design in Experiment 3.

- Were the models fit separately for valence and arousal? Or have you tried a multivariate fit/distribution? From Line 241, it sounds like it is the former. In that case, I think separate results should be presented at least as a supplement, so the reader could see the relative contribution of each dimension. Without this information we do not know whether the superiority of the simple averaging model exists in both dimensions or is it due to for instance better performance on one dimension and a generally inadequate fit of all models in the other.

We now provide valence and arousal performance for all experiments as supplemental figures (Figure S2).

Figure S2

We find that the Heuristic model outperformed the Bayesian model in both valence and arousal dimensions in Experiment 1. In Experiment 2, we find no difference in performance between the Bayesian and Heuristic models in the valence dimension ($t(67) = 0.52, p = 0.603$) nor in the arousal ratings ($t(67) = 1.87, p = 0.07$). Finally, we find a significant difference in performance between the Bayesian and Heuristic models when predicting observers' ratings in valence ($t(113) = -6.77, p < 0.001$) but not in arousal ($t(113) = -0.44, p = 0.662$) dimensions in Experiment 3.

We have added these results in the supplementary information (Figure S2).

- I think the authors should also consider a weighted averaging model in which the cues are weighted instead of simply averaged. It would be interesting to see if the weight for the character-only cue is greater than the context for static images (randomly presented) in which there is not much of an evolving context.

We thank the reviewer for the alternative model suggestions, we provide a new analysis which weighs the character-only cue 75% and the context-only cue 25% and vice-versa. We call these models the Character75 and the Context75 models. This allows us to

investigate model performance when overweighting one cue versus the other. We have added the results to lines 939-1043 in the manuscript. We found that these models with stable weights did not outperform the Heuristic or Bayesian model in all Experiments (Figure 7).

Figure 7

- In introduction, discussion and even in abstract, the subject of the study is formulated as the integration of emotional information from facial cues and context. However, the stimuli in

character-only condition involve much more than facial expressions. They also include posture, body language, and movement. This is mentioned in passing but I think it should be clearly stated from the abstract what different cues represent. I think this is also one of the main reasons that the results differ from for instance from Goel et al., 2024 (reference 25).

We recognize that our current framing of the manuscript focuses on facial cues when what we really mean is face and body cues. We solely focus on facial cues in the introduction for introducing the research question at hand, however we make sure to refer to face and body information throughout the rest of the manuscript.

- Please present correlations between context- and character-only ratings as a supplement.

The analysis has been added to the supplemental information (Figure S3).

- The methods are not described very clearly. At times I needed to go and read the method sections of Ortega, Chen, Whitney, 2023 & Chen Whitney 2019 to understand the procedure. The explanations of the procedure in Methods section (from Line 184) are confusing. This entire section should be revised to clearly present the procedures. To give a concrete example, the description of Experiment 1 (Line 184-209) does not make it clear that the participants did not view any video stimuli but they instead only rated still images presented randomly. The paragraph starts explaining that participants viewed independent stimuli "Each observer was presented with shuffled frames from different videos presented in a random order" (Line 190). But then it continues "Observers in the context-only condition continuously tracked the emotions of a target character who was blurred out in the video clip." (Line 193). Please go over the entire method section and improve the clarity.

We have revised the methods section so it is more concise and clear.

- Comparison for Pearson r at Line 387-388. I was surprised to see that these are different and yet both so close to ceiling. What is the unit of comparison in the t-test with $df=67$?

We thank the reviewer for pointing out this discrepancy; we have clarified and corrected the results section. Our original analysis code had an error and we have corrected it. The fixed code does indeed reveal that the Heuristic model is outperforming the Bayesian model, except for when calculating AIC (Figure 5):

We have revised the results section to reflect these results.

Minor comments and typos:

- No age and gender distributions are presented for the sample in Experiment 2.

Data for Experiment 2 in our study was obtained from a subset of the data in Chen & Whitney (2019). We do not have access to individual observer demographics for this study and only have access to the full dataset distribution of age and gender.

- Line 170-171. "In Experiment 1, there was a total of 593 participants." repetition of the previous sentence.

The repeated sentence has been deleted.

- Please clearly state whether Experiment 3 is a new data collection for this manuscript

We have added: “Unlike Experiment 1 and 2, the data collected for Experiment 3 was novel and only used in this study.”

- Please clearly state which experiments are Experiment 2a and 2b in Chen & Whitney, 2019.

We have added: “ In Experiments 2a and 2b, data was retrieved from Chen & Whitney (2019)¹ which included data from Experiment 2 and Experiment 3 in their study.”

- Eq2 (LSE function). What purpose this function fulfills should be explained for the reader?

To explain LSE, we have added “*This allows for normalizing $P(a|c)P(a|f)$ by subtracting the sum of the log probabilities of $P(a|c)$ and $P(a|f)$ instead of dividing by $P(a)$.*” to the methods section.

- Line 294. It should be “when the log probabilities are available”

We have corrected this sentence.

- No mention of sounds? Were the video stimuli silent?

The videos were silent and had no sound. We have added this information in the methods section.

Reviewer #2 (Remarks on code availability):

I have not tried to install or run the code. I only looked at the code and it seems that it checks out.

Reviewer #3 (Remarks to the Author):

This paper examines how people integrate emotional information from facial expressions and context across three studies with different dynamics, highlighting a heuristic integration model as the most parsimonious model over a full Bayesian model. This integration model averages cue information without considering cue ambiguity, which makes it less computationally costly. The authors presented their results clearly and while I am interested in understanding how people integrate emotional information, I have major concerns about assumptions from the Bayesian model that might need to be accounted for to properly conduct model comparison. I also would like the authors to investigate potential differences in model fits across emotional video stimuli, something I think that might reveal a potentially interesting and nuanced understanding of how people integrate these cues.

Major concerns

1. My primary concern centers around the assumptions behind the Bayesian integration model. Specifically, the model assumes that each cue is represented with a normal probability distribution (mean and standard deviation). The authors do not present any evidence that the affect ratings are properly represented by a normal distribution and if this assumption is inappropriate, it could cause the Bayesian model to perform poorly. It's quite likely that affect ratings are not always normally distributed; the supplement of one of the published datasets (Chen & Whitney, 2017) shows 2D distributions that do not look like a multivariate normal. I recognize that the authors simply used the formulation from prior research (Ong et al., 2015), which is completely understandable. However, I think in this context where authors want to argue for the superiority of the heuristic model, that the potential violation is highly relevant.

We thank the reviewer for the insightful comment. However, we would like to point out that our modeling approach does not involve using the 2D distribution of valence and arousal (as shown in Chen & Whitney, 2019). Instead, we model observers' responses either by trial-by-trial (Experiment 1) or timepoint-by-timepoint (Experiment 2 and 3) based on the ratings across all observers (refer to Figure 4 in the manuscript). If we were interested in modeling the average rating for all videos in our dataset, then the 2D distribution would be important to consider.

2. A more complete model comparison could include a non-parametric Bayesian model. The authors have a large amount of data, which should make it ideal for estimating the empirical distribution using kernel density estimation or other non-parametric tools. This would allow a non-parametric Bayesian integration model to more properly reflect the empirical priors, rather than collapsing potentially non-parametric distributions into a mean and variance rating. Even if the affect distributions are normal, this modeling approach could simply approximate the normal and offers a more flexible model to account for the current and future data that extend beyond the situations captured in the emotional video stimuli. The heuristic model effectively just takes the maximum likelihood estimate, making it robust to distribution concerns and is potentially one explanation for why its performance is superior stimuli which might differ in their distributions. This is also could be framed as a positive for the heuristic model since it is more agnostic to distribution.

We have now included a section in the supplementary that uses kernel density estimation to fit distributions on the data:

Figure S4

We find that fitting adaptive-KDE distributions to the data led to worse performance than the original Bayesian model in both Experiments 1 and 2. Experiment 3 data was not analyzed with KDE distribution since it had a within-subject design and observers only rated the videos once and not multiple times. The KDE result may seem surprising, however, it suggests that observers may not be estimating the exact distribution of each cue as it would increase the computational load needed to integrate cues. (c) Interestingly, we do find that some videos ($n=8$) are best modeled by the adaptive KDE model suggesting that observers may try to estimate the true distribution of cues during certain conditions.

The figure has been added to the supplementary information as Figure S4.

3. Another major concern is that the authors chose to model valence and arousal separately, without accounting for the covariance between the two. Emotion research has shown a group-level quadratic-like relationship between valence and arousal (Kuppens et al., 2013) and this is particularly important for modeling individual variability in how valence and arousal are related at the individual level. I would like to see the authors apply a joint integration model, especially since the valence-arousal ratings are not independent but averaging across those models effectively treats them as independent. This may be more difficult for the proposed non-parametric Bayesian model, but doable for a multivariate normal Bayesian model.

While previous research has shown that valence and arousal have a U shaped relationship at a global level when taking into account the full gamut of possible affective states, this relationship may not hold at the local level, within any given individual dynamic scene. Indeed, figure R4a below shows that each video has affect ratings that only occupy a local region in valence and arousal space, and those local regions are, in fact, independent. Previous studies (Kuppens et al., 2013) primarily investigated ratings of random, unrelated static stimuli (the global affect space), but the relationship between valence and arousal in natural dynamic ratings have yet to be investigated. We explored this question in our data by plotting the valence and arousal ratings for the dynamic ratings (Figure R4b):

Figure R4

In order to investigate if there was a U shape in our ratings in local space, we mean differenced all ratings within our videos and then overlaid them on top of each other (Figure R4b). We also plotted the best fit quadratic function (r -squared: 0.002). We find that when looking at valence and arousal ratings at the local level, no relationship is observed.

4. Relatedly, I worry about averaging model performance across valence and arousal. It is understandable for ease of communication, but it's likely the integration of these signals differs. For example, valence is recognized as a main dimension of emotion across cultures while arousal ratings show stronger cultural differences (Jack et al., 2016). Also, a recent large dataset examining valence and arousal ratings shows differences in their distributions even within the same emotion categories (Heffner & FeldmanHall, 2022). Given this I might predict that overall, the valence integration models will fit better than the arousal integration models. It would be helpful if authors separated model performance for valence and arousal, at least to report in the Supplementary materials.

We have now analyzed valence and arousal performance separately for all experiments, presented as supplemental figures (Figure S2).

Figure S2

We find that the Heuristic model outperformed the Bayesian model in both valence and arousal dimensions in Experiment 1. In Experiment 2, we find no difference in performance between the Bayesian and Heuristic models in the valence dimension ($t(67) = 0.52, p = 0.603$) nor in the arousal ratings ($t(67) = 1.87, p = 0.07$). Finally, we find a significant difference in performance between the Bayesian and Heuristic models when predicting observers' ratings in valence ($t(113) = -6.77, p < 0.001$) but not in arousal ($t(113) = -0.44, p = 0.662$) dimensions in Experiment 3.

We have added these results in the supplementary information (Figure S2).

5. I think there's more nuances hidden in the data beyond Bayesian vs Heuristic, and I would love to see the authors present some of the stimuli-specific comparisons. For example, it would be very interesting to see if any of the emotion video stimuli the Bayesian model greatly outperforms the Heuristic model. The Figure 4 visualization intentionally suggests this as the Bayesian integration moves from character to context throughout the video. It was not clear to me how many of the emotional stimuli are shared across the experiments, but assuming there's enough overlap then the authors have an ideal chance to show out of sample generalization (e.g., Bayes model for video 4 beats heuristic model in all three experiments). This could really

elevate the understanding of when and why people may want to incorporate the cue ambiguity. Alternatively, if the Heuristic model almost always wins at the emotion-stimuli level this would strengthen the authors conclusions, but if the winning model is inconsistent within stimuli across experiments this would weaken the conclusions from model comparison. It seems plausible that the integration of cue ambiguity might change with the video's context and would fit with a large literature on situation-specific emotion appraisals.

We investigated model performance based on each individual video for performance and the results can be seen in the following figure:

Figure S5: (a) and (b) compare AIC values between the Bayes and Heuristic model for Experiment 2 and Experiment 3, respectively. The distribution in Experiment 2 was not significantly different from 0 ($\chi^2 = 2.12$, $p = 0.1456$) but the distribution for Experiment 3 was ($\chi^2 = 6$, $p = 0.0143$) indicating that the Bayesian model outperformed the Heuristic model in Experiment 3 at the individual video level. Additionally, Figure (c) shows the same data as Figures a and b but for the 12 videos (for both valence and arousal ratings) that overlapped between Experiment 2 and 3. We compute the AIC difference between the Bayes and Heuristic models, where positive values indicate that the Heuristic model is better while negative values indicate the Bayesian model was better. We find a significant positive correlation between model performance differences for videos present in both Experiment 2 and Experiment 3 indicating that model performance may generalize across experiments for the same video (Spearman $r = 0.41$, p -value = 0.022).

The current figure has been added to the supplemental information (Figure S5).

6. The authors should report protected exceedance probabilities (PXP) which is a measure of how likely any given model is compared with all other models in the comparison set (e.g., Piray et al., 2019; Stephan et al., 2009 for original). This also would not require averaging across valence / arousal or emotional stimuli since those could be treated as “subjects” typically are in these modeling papers. This is particularly important and different from comparing lowest average AIC as it considers the relative performance of all models in the set. Because the Bayesian and Heuristic models have similar performance, this measure is ideal to account for model uncertainty and identify the model that is most likely to be the best among the set.

We have calculated PXP values for Experiment 3 and find that PXP measures also agree with Experiment 3 results:

The Bayesian model had higher PXP values than the Heuristic model, indicating that it better fit observers' ratings in Experiment 3 than the Heuristic model. We have added this analysis to the supplemental (Figure S6).

7. Relatedly, I would like to see individual difference measures in Experiment 3 for model fits. For example, a scatter plot of the AIC values for Bayes vs Heuristic model per participant as well as the percentage of subjects who are best fit by either model. One possibility if the data are scattered close to the identity line, it suggests about ~50% are best fit by either model and that participants are essentially indifferent between which model best fits their data. To me, this would suggest that neither model is the “true” model and would offer additional nuance to how people integrate across these cues at the individual level. If this plot reveals clear differences (e.g., 70% of participants are best fit by the Heuristic model), then this strengthens the authors claim for Exp 3 since the Heuristic model ought to be preferred if it fits most participants.

Good point; we agree. To address these comments, we examined the individual differences testing whether individual observer's ratings were better captured by the Bayesian model than the Heuristic model as seen below in Figure S1:

We investigated whether there were individual differences in integration strategies by plotting model performance between the Bayesian and Heuristic models for each individual participant. We first computed the differences between the Akaike information criterion (AIC) values of the Bayesian model and the Heuristic model for each participant. Positive values indicate that the participants' data was best captured by the Heuristic model while negative values indicate the Bayesian model best captured their ratings. We then compared the AIC difference scores to a permuted null distribution. The permuted null was computed by shuffling observers' ratings such that a random observer's context and character ratings were integrated to predict each observer's ground truth rating. This allows us to investigate whether there are significant idiosyncratic differences within observers, or if observers' data is interchangeable between observers.

(a) shows an example of the AIC difference scores obtained for a single observer's permutation (gray distribution; 100 iterations) compared to their observed AIC difference score (vertical red dashed line).

(b) shows the same analysis for all observers where the shaded region is the 95% confidence interval (CI) of the permuted null distribution. Observers who had significant

AIC differences (those that were outside of the permuted 95% CI) are shown as a yellow diamond and have an asterisk (*) at the top of the figure. In total, there were 30 observers who had significant individual differences which is significantly more than expected by chance ($\chi^2 = 101$, $p < 0.001$). Importantly, this suggests that observers cannot be swapped with other observers, indicating significant individual differences.

(c) We also found that more individual observers were best modeled by the Bayesian model than the Heuristic model ($\chi^2 = 16.98$, $p < 0.001$).

(d) To further explore whether there were significant individual differences within observers, we performed a split-half analysis where we split each observer's AIC difference scores in half, randomly, and compared the average AIC difference score for each half for 10,000 iterations using RMSE (within-subject analysis). We compared this to a between-subject analysis where we randomly selected a pair of observers AIC difference scores and compared the average AIC difference score for 10,000 iterations using RMSE (between-subject analysis). We find that AIC differences are more similar within-subjects than between-subjects, providing further evidence for individual differences among observers ($p = 0.0072$; permutation test).

(e) Finally, we investigated how the performance of the Bayesian model differs if we were to shuffle context and character ratings across observers. We find that integration cues retrieved from a random pair of observers lead to worse model performance than when using each observer's own ratings for character and context cues ($t(113) = -12.72$, $p < 0.001$).

Overall, these findings suggest that different observers may use different integration strategies when combining affect information, highlighting the presence of idiosyncratic differences across observers.

These results have been added as supplementary Figure 1 (Figure S1).

Minor concerns

1. It would be very helpful if authors were to show the empirical distributions for valence and arousal ratings for each emotional video stimuli, this could go in the Supplement.

We believe that showing the empirical distributions for valence and arousal for each video stimulus will mislead readers to think that these distributions were used in our modeling approach. Instead, we provide the distributions for Experiment 1 here for the reviewer as an example (Figure R3):

Figure R3

Again, we would like to reiterate that these distributions show the valence and arousal distributions across the entire video. These distributions were not used in our modeling approach, as we modeled valence and arousal for each time point in the video.

2. The authors use the term “emotion rating” in many places where “affect rating” is more appropriate. For example, Figure 2 the x-axis is really the affect ratings (valence or arousal). In the computational models, they say they’re modeling judgments of emotion, but again valence-arousal ratings are more commonly referred to as affect. I would recommend changing these to distinguish this work from work that is specifically about inferring or judging emotions (i.e., “angry”, etc.).

We have changed *emotion* to *affect* in our manuscript.

3. Figure 5a. The Pearson correlation bar for Bayes is visually higher than the Heuristic model, but in text the authors report that the Heuristic model has a larger correlation than Bayes (Heuristic r 0.917 and Bayes 0.914, lines 384-390). Please confirm and check.

Thank you for pointing out this discrepancy. This was due to an error in our graphing code, we have updated the figure to the correct version.

4. Figure 6a could be clarified. I do not understand the output rating graph, is this meant to reflect the affect rating on the x-axis and the cumulative PDF on the y-axis reported from the model? The authors might want to label the y-axis since the model figure uses a PDF. I expected to see a normal distribution for the output rating.

The output rating graph is meant to represent the dynamic rating of the full video for each model. We have added *Time* and *Rating* to the x and y axis, respectively, to make it clearer.

5. Figure 6a: it might help to have multiple frame examples (i.e., stacked frames) to visually indicate this was for continuous movies.

We have added stacked frames to Figure 6a.

6. The citations for Goel can be updated into the recent published version (Goel, Jara-Ettinger, Ong, & Gendron, 2024, Nature Communications).

The citation has been updated.

Reviewer #3 (Remarks on code availability):

I did not review the code line by line but skimmed it to understand how their approach might be able to incorporate a non-parametric Bayesian model.

I think the authors could include more details in the README or comments of their markdown to make it easier for future readers unfamiliar with these models to follow, but this is also a high bar that is not currently the norm in the field.

We thank all reviewers for their constructive comments. We have addressed all comments below in the bolded text and have also highlighted parts of the manuscript that were added in response to reviewers' comments. We have also recognized the discussion section and included section titles.

REVIEWER COMMENTS

Reviewer #1 (Remarks to the Author):

We thank the reviewer for their thoughtful comments and for reading through the manuscript again. We provide responses to each point below.

I was Reviewer 1 on the previous round of review. The authors have done a commendable job at answering all my and the other reviewers' comments, including reporting many more analyses.

In general, I'm not surprised by many of the "new changes" in the results, including that (a) the simpler Heuristic model does better for Expt 1 and 2, while (b) the Bayesian model does better at within-subject predictions in Expt 3, and that more observers' behavior are better predicted by the Bayesian than the Heuristic model, but that there exists substantial individual variability. This jives with my intuition, and with previous work (e.g., Goel et al., 2024), and I think this paper makes an exciting contribution to the literature.

I also appreciate that the authors have tempered down their inferences, and I found the additional analyses (e.g., the graphs in the revision letter and supplemental) very helpful in clarifying the nuances in the results.

- I do have a minor question though: for the new KDE-estimation analysis, isn't the normal Gaussian model recoverable by KDE? That is, if the normal Gaussian is really the best performing model, then KDE should be able to recover that normal Gaussian as it's a point in the model space that the KDE will explore. I'm surprised that the KDE does so much worse (comparable to and even worse than the character-only model). I suppose it could be because of the group-level estimation.

The Gaussian model would be recoverable by KDE if the rating distribution is Gaussian. The KDE-estimation model fits KDE distributions to the data and then uses these distributions when integrating the two cues instead of forcing a Gaussian distribution during integration. The fact that the model performs worse may suggest, as you mention, that individual differences may be playing a part, which would decrease model performance when using group-level estimation. We have now acknowledged this in the discussion section:

“Alternatively, lower accuracy by the kernel density estimation model could be due to the presence of individual differences. If there are individual differences in affective judgments or even in the estimation of ambiguity in the cues, then this could lead to worse model accuracy since accuracy is determined at the group-level, and not the individual level for Experiments 1 and 2.” (lines 684-688)

I think that this paper is much improved, and I only have minor comments left. I think the writing can be improved in many places, and a thorough proof-read would improve the paper. Here is a non-exhaustive list:

“To conclude, we found that the integration of affective cues in context-rich, dynamic scenes follows a Bayesian framework if individual observer integration is taken into account.” – this last clause is unclear... I’d suggest rewriting this.

We have now changed this section to:

“To conclude, we found that the integration of affective cues in context-rich, dynamic scenes follows a Bayesian framework. However, different observers can use different integration strategies when combining different sources of information, highlighting the presence of idiosyncratic differences across observers in emotion perception.” (lines 834-837)

“Thus, to accurately infer affect, one needs not only facial expressions and body language, but context as well.” – the claim in this paragraph-final sentence does not follow from the entire paragraph before. The paragraph is talking about multiple model comparisons and the weightage of different cues.

Thanks for catching this, we have removed this sentence as it is already mentioned in the first paragraph of the discussion.

“Optimal integration is needed to accurately perceive the affect of others” It’s difficult to make claims about affect perception “accuracy” because accuracy usually implies some concordance with the “truth”. In affective science this is usually taken to be the target’s self-report (as we lack better ways of measuring truth), and an observer’s perception or inference is said to be accurate iff it matches the target’s self-report. In all of the experiments, you do not have the target’s actual self-report, so I would argue that accuracy is overclaimed here. Apologies that I didn’t catch this in the initial submission. I will note that most of the uses of “accuracy” in the paper is used to mean “model accurately predicting human judgments”, which is an accurate use of the term (pun intended), but there are a handful of places where “accurate” (in “accurate affect perception”) is imprecise.

We agree, we have now changed the term “accuracy” to only reflect model accuracy and not perceptual accuracy. For example, in the mentioned sentence, we now state:

“Optimal integration is needed to model human judgments of affect” (line 497)

“Ground truth” as used in the paper is not used in the commonly used sense of a true “label”. In this paper, “ground truth” means “all cues are observed”.

We use the term “ground truth” because it is what is being used to compare model outputs and measure model performance. While it is not being used in its conventional manner where it usually refers to the true emotions of a person, we still believe that it is an appropriate label since it is how we judge whether a model is accurate. We have added the following to the manuscript to make this clear:

“This condition is termed as the “ground truth” because it is what will be used to compare model outputs and estimate model performance.” (lines 195-196)

Reviewer #3 (Remarks to the Author):

I want to thank the authors for their comprehensive response and completing many additional analyses. All of my prior concerns have been addressed and the authors have modified their conclusions regarding the strength of the Bayesian vs Heuristic account accordingly.

After reviewing the revised manuscript, I have no further concerns.

We thank the reviewer for their constructive comments!

Reviewer #4 (Remarks to the Author):

The authors investigate an important question in emotion attribution: how humans integrate information about an agent’s face and body with information about the surrounding environment during emotion inferences. They do so using frames from videos, leveraging two innovations: the use of naturalistic stimuli, and dynamic sampling of emotion ratings during the course of a video. The authors investigate integration between face/body and context at the group level, as well as individual differences. The study of individual variation is an important topic that deserves more attention in the field – I praise the authors’ effort to engage with this challenging area of research. In my understanding, the analyses rest on a set of assumptions. One of these assumptions – that uncertainty distributions are normal – has been addressed in a response to another reviewer. However, the two other assumptions (discussed below) would need to be tested in order to support firm conclusions about cue integration. I see that the authors already did an impressive amount of work responding to an initial round of reviews, and I believe that it would be useful for the field to have this work in published form. At the same time, the untested

assumptions moderate the conclusions that can be derived from the results. At a minimum, these assumptions should be thoroughly discussed in the Discussion section.

We thank the reviewer for their thoughtful comments. We provide responses to each point below.

Major:

1. According to the methods section “Observers in the context-only condition rated the affect of a target character who was blurred out in the frame.”. This suggests that participants were asked to provide a single response for the affect of the person in the video frame, rather than a probability distribution describing how likely the participants thought different affects were. Previous work highlighted the difference between the distribution of ratings across participants and the distribution describing uncertainty within individual participants, finding that the two do not always match. Could this affect the study’s conclusions? Individuals who appeared to be better described by the heuristic model might instead be participants whose uncertainty distribution has a minor mode, that was not captured due to asking participants to generate a single affect judgment at each moment rather than asking them to generate a probability distribution that describes their uncertainty. Vice versa, it might also be possible that asking participants to generate a probability distribution describing the likelihood of each possible value of affect (rather than a judgment about the most likely affect) would have revealed more shortcomings of the Bayesian integration approach. The conclusions rest on the assumption that the distribution across participants is equal to the uncertainty within participants. This assumption needs to be tested, or at a minimum this must be acknowledged as a limitation of the study.

This is a good point. Recent work has shown that model performance in the integration of affective ratings is modulated by the methods in how observers report their emotional judgments (Anzellotti et al. 2021). We now acknowledge this limitation and cite this study in the discussion section. Based on this, model performance might change if we asked participants to report the probability distribution of affect instead of a single continuous judgment. However, implementing this method would prove difficult or impossible in our study design. Having observers report a probability distribution that describes their uncertainty of the target character's emotion *continuously*, would greatly increase the difficulty of the task for observers, if it's even possible. There is no clear way to implement probability distribution reporting in our task **since participants would need a few seconds to report this and this isn't really feasible continuously in two dimensions. Alternatively, we could have participants watch and rate the same videos multiple times (and potentially across multiple days). However, this would lead to further confounds, including exposure effects, learning, prediction (known future), and other unknown interactions that can't be controlled.**

We have added the following limitation to the discussion section to address this concern:

“Another limitation in our study is that we did not directly measure individual observers’ estimates of cue ambiguity and instead used group-level estimates of ambiguity for modeling. Recent work has shown that model performance in the integration of affective ratings is modulated by the types of methods used to measure how observers report their perceived ambiguity of emotional judgments (Anzellotti et al. 2021). As we observed in our study, prominent individual difference in affective cue integration exists, and we should expect that this should impact observers’ estimates of cue ambiguity uniquely for each individual.” (lines 810-817)

2. In Experiment 3, the authors write: “We used the observer’s empirical rating at each time point as the mean of the normal probability distribution for each cue. We also used the standard deviation of the ratings across all observers for the normal probability distribution of each cue.”

If the probability distribution for each individual observer was modeled using the standard deviation across all observers, this would lead to an incorrect estimate of the distribution of the uncertainty within an individual unless three assumptions are met: first, the assumption already discussed in point 1; second, the assumption that the uncertainty within individuals follows a normal distribution; and third, the assumption that individual differences only affect the mean and not the standard deviation. The issue arising from the second assumption has been addressed using KDE. However, the other two assumptions have not been tested. It is possible I am misunderstanding something, but if my understanding is correct, deriving firm conclusions about cue integration will require testing these assumptions.

We directly responded to the first assumption above and have addressed the second assumption with our KDE model as the reviewer mentions. As for the third assumption, we completely agree – their should be individual differences in the ambiguity (i.e. the standard deviation of the cue distribution). We now acknowledge this in the added discussion section in response to point 1 above:

“As we observed in our study, prominent individual difference in affective cue integration exists, and we should expect that this should impact observers’ estimates of cue ambiguity uniquely for each individual” (lines 814-817)

3. As noted by Reviewer 1, the word “context” can take on different meanings. In some cases – as in this study – “context” refers to the environment surrounding a target agent. In other cases, “context” is used to refer to situations that lead up to an observed event (for example, “Mark stole a piece of bread after he ran out of food to feed his family.”) I read the authors’ response to Reviewer 1 on this point, and I still think that context in the sense of “the environment surrounding an agent” and context in the sense of “the representation of the situations leading up to the current frame” are different in general. In some cases, information about the environment surrounding an agent in a video frame can be used to make inferences about the situations that led to the current frame, but this will vary case-by-case. Event understanding and semantic knowledge might play different roles in the analysis of the two types of “context”. This

does not make the current manuscript less interesting: investigating the role of information about the surrounding environment is valuable as well. However, I concur with Reviewer 1 that this distinction should be made explicit when the study is compared to previous studies that used the term “context” in a different way, because clarifying this will help prevent potential confusion from the readers.

Thanks for this point. We agree that temporal context (e.g. “the representation of the situations leading up to the current frame”) is different than the context of the agent’s surrounding environment. The former is temporal context while the latter is spatial context. We still consider both as context because in our study, the target character (or Mark in your example) is completely blurred out at every time point, which means that the only available information comes from outside of the agent. As mentioned, this includes both spatial (e.g. surrounding environment) and temporal (e.g. event understanding, etc) context.

We have added the following to the discussion to address this point:

“Integration for spatial context (e.g. a person’s surrounding environment) and temporal context (e.g. chronological narrative information) may also be integrated at different time scales.” (lines 703-705)

“Future studies should also investigate and try to differentiate whether cue integration varies for different kinds of context (e.g. spatial vs temporal context).” (lines 825-827)

Minor:

4. While the total number of frames is large, the number of videos used is relatively small (34). It is unlikely that the range of scenarios is representative of the variety of situations that occur in daily life. I appreciate that the authors mentioned this in the Discussion section.

We agree and have included the limitation in our discussion section.

Reviewer #4 (Remarks on code availability):

The code looks clear and straightforward to use to reproduce the results.